# Discovery of NSD2 non-histone substrates and design of a super-substrate
Sara Weirich, Denis Kusevic, Philipp Schnee, Jessica Reiter, Jürgen Pleiss ⓘ & Albert Jeltsch ⓘ ✉

The human protein lysine methyltransferase NSD2 catalyzes dimethylation at H3K36. It has very important roles in development and disease but many mechanistic features and its full spectrum of substrate proteins are unclear. Using peptide SPOT array methylation assays, we investigate the substrate sequence specificity of NSD2 and discover strong readout of residues between G33 (-3) and P38 (+2) on H3K36. Unexpectedly, we observe that amino acid residues different from natural ones in H3K36 are preferred at some positions. Combining four preferred residues led to the development of a super-substrate which is methylated much faster by NSD2 at peptide and protein level. Molecular dynamics simulations demonstrate that this activity increase is caused by distinct hyperactive conformations of the enzyme-peptide complex. To investigate the substrate spectrum of NSD2, we conducted a proteome wide search for nuclear proteins matching the specificity profile and discovered 22 peptide substrates of NSD2. In protein methylation studies, we identify K1033 of ATRX and K819 of FANCM as NSD2 methylation sites and also demonstrate their methylation in human cells. Both these proteins have important roles in DNA repair strengthening the connection of NSD2 and H3K36 methylation to DNA repair.

Together with DNA methylation and the presence of non-coding RNAs, histone posttranslational modifications (PTM) regulate many chromatin-templated processes[1], control cellular phenotypes, and the development of diseases[2,3]. The most prominent histone PTMs are acetylation, phosphorylation, ubiquitination, and methylation[4]. Histone lysine methylation is an extensively studied modification and up to three methyl groups can be transferred on lysine residues by specific Protein lysine methyltransferases (PKMTs)[5,6]. Depending on the site and degree of methylation, effector proteins with specific binding properties are recruited and regulate further downstream biological processes[7]. Besides the methylation of lysine residues of histone tails, high-throughput proteomic studies led to the identification of lysine methylation at numerous non-histone proteins where it controls important processes, like protein degradation, protein localization, and protein-protein interactions, which in turn regulate many biological processes[8–10]. Up to date, the knowledge about non-histone substrates is incomplete for many PKMTs and it is often difficult to associate the matching substrates and PKMTs. One approach to search for possible non-histone substrates of PKMTs is to analyze their substrate specificity and use this information to identify novel substrate candidate proteins[11].

The Nuclear Receptor Binding SET Domain Protein 2 (NSD2, also known as MMSET, or WHSC1), together with NSD1 and NSD3, forms the nuclear receptor SET domain-containing (NSD) enzyme family. The NSD enzymes are key epigenetic enzymes that catalyze H3K36 mono- and dimethylation[12–14]. H3K36 methylation in gene bodies is associated with active transcription and splicing and this modification is also involved in the regulation of heterochromatin formation, DNA replication, recombination, and DNA repair[13–15]. The NSD enzymes differ in protein length, NSD2 with 1365 amino acids (aa) is the smallest among all NSD family members. All three NSD enzymes share a SET domain as catalytically functional part, flanked by AWS (Associated with SET) and Post-SET domains. In addition, they contain PWWP (proline-tryptophan-tryptophan-proline motif) domains and PHD (plant homeodomain) domains, which are important for mediating interaction with chromatin and other proteins[16]. In contrast to all other NSD family members, NSD2 also contains a high mobility group (HMG) domain that interacts with the DNA binding domain of the androgen-receptor (AR) and results in enhanced nuclear translocation of both proteins[17]. The gene encoding NSD2 was identified in 1998 when it was observed that patients suffering from the Wolf-Hirschhorn syndrome have either a partial or complete deletion of the NSD2 gene, which leads to a haploinsufficiency of NSD2, resulting in the name Wolf-Hirschhorn syndrome candidate 1 (WHSC1)[18]. NSD2-deficient mice exhibit phenotypes similar to the human WHS, such as growth defects, deficiencies in midline fusion, and congenital heart defects, and they die 10 days after birth[19].

Institute of Biochemistry and Technical Biochemistry, University of Stuttgart, Allmandring 31, 70569 Stuttgart, Germany.
✉e-mail: albert.jeltsch@ibtb.uni-stuttgart.de

Moreover, the Wolf-Hirschhorn syndrome is linked to defects in the DNA damage response[20].

Later, it was discovered that NSD2 has PKMT activity and different studies reported that NSD2 catalyzes dimethylation of H3K36[19], H3K4, and H3K9[17], trimethylation of H3K27[21], di- and trimethylation of H4K20[22,23], and monomethylation of H4K44, but this reaction was not observed in the chromatin context[24]. However, there is disagreement among published reports regarding the methylation activities of NSD2 in particular at H3K4, H3K27, and H4K20[24]. In addition, several non-histone substrates of NSD2 with important roles in tumorigenesis and tumor progression have been described. Song et al.[25] detected that lysine 163 methylation of Signal transducer and activator of transcription 3 (STAT3) by NSD2, leads to an activation of the STAT3 signaling pathway, which promotes tumor angiogenesis[25]. Methylation of the Aurora kinase A (AURKA) by NSD2 was discovered, which reduces P53 stability resulting in increased cell proliferation and oncogenic activity[26]. NSD2 has been shown to be crucial for the recruitment of the P53 binding protein (53BP1) to DNA double-strand breaks[23]. Later, NSD2 was also shown to methylate K349 of Phosphatase and tensin homolog (PTEN) K349, which is bound by the Tudor domain of 53BP1 and facilitates its recruitment to DNA double strand breaks[27].

In addition to its key role in WHS, many reports demonstrated that NSD2 is mutated in several tumors[28,29]. Especially, the E1099K mutation, located in the active site pocket in a loop next to the bound substrate, is recurrently occurring in lung adenocarcinoma where it increases NSD2 activity promoting KRAS signaling and other oncogenic gene expression programs[30]. The effect of the hyperactivating mutations is comparable to the chromatin alterations often leading to NSD2 overexpressing in tumor cells[29]. The activating T1150A mutation in NSD2 is often found in leukemia patients[31]. This mutation also changes the product specificity from H3K36me2 to H3K36me3[32].

In this study, we investigated the peptide sequence specificity of NSD2 and discover strong readout of residues between G33 (-3) and P38 (+2) on the H3K36 sequence. However, we observed that amino acid residues different from the natural ones in the H3 tail were preferred at some positions in the substrate peptide. Combining four of these preferred residues led to the development of a super-substrate (ssK36) which was methylated about 100-fold faster by NSD2 than H3K36 at peptide level and even more preferred at protein level. Molecular dynamics simulations demonstrated that this activity increase is caused by distinct hyperactive conformations of the enzyme-peptide complex which are adopted if ssK36 is bound. In addition, based on the specificity profile of NSD2, we identified K1033 of ATP-dependent helicase (ATRX) and K819 of Fanconi anemia group M (FANCM) protein as NSD2 protein substrates in vitro and demonstrated their methylation in cells. Methylation of these proteins by NSD2 has not been shown before to the best of our knowledge. Both these proteins have important roles in DNA repair and they strengthen the connection of NSD2 and H3K36 methylation to DNA repair.

## Results
### Substrate specificity analysis of NSD2
Several previous publications reported that PKMTs methylate different substrates, but for many enzymes of this class, the full substrate spectrum is not known[11]. Knowledge about PKMT substrates is essential to understand their biological roles because each individual protein methylation event can regulate a specific biological pathway[6,8–10]. One approach to determine the full substrate spectrum of PKMTs and identify novel substrates is to characterize the sequence specificity of the enzyme and use this information to identify novel potential substrate proteins[11]. For this task, peptide SPOT arrays can be employed, as they allow to study the methylation of several hundred peptides in one experiment at moderate costs[33,34]. NSD2 is an essential PKMT which generates H3K36 mono- and dimethylation and has essential roles in chromatin regulation, cell physiology and cancer biology[12]. To investigate the substrate specificity of this enzyme, we have cloned the catalytic SET domain (aa 991–1240) of human NSD2 (Uniprot O96028) in a GST-tagged form, overexpressed it in *E. coli* cells, and purified the enzyme

by affinity chromatography with good yield and quality (Supplementary Fig. 1a). To confirm its methyltransferase activity, peptide arrays containing 15 aa long peptides of potential histone substrates and the corresponding K-to-A mutations as negative controls were synthesized on a cellulose membrane using the SPOT technology. The peptide arrays were incubated with NSD2 in the presence of radioactively labeled AdoMet as cofactor. As expected, the autoradiographic image confirmed the methylation of H3K36 and loss of methylation for the corresponding negative control (Supplementary Fig. 1b). In addition, we detected previously reported methylation of H4K44[24] and strong methylation of H1.5K168, which was previously identified as preferred substrate of NSD1[35]. In all these peptides, methylation occurs in a G V **K** (KR) ψ sequence context (ψ represents a hydrophobic amino acid residue). To test the PKMT activity of NSD2 with protein substrates, recombinant H3.1 and H4 were incubated with NSD2 in the presence of radioactively labeled AdoMet. Since many PKMTs show an automethylation activity[36–43], an additional sample without external protein substrate was included. After methylation, the samples were separated by SDS-PAGE and methyl transfer was detected by autoradiography. In this experiment, methylation of recombinant H3.1 and H4 was confirmed and in addition, a strong automethylation signal of NSD2 in absence and presence of recombinant H3.1 and H4 was detected (Supplementary Fig. 1c).

We next intended to investigate the detailed substrate sequence specificity of NSD2 using H3K36 as template sequence. Therefore, three peptide SPOT arrays were synthesized, in which each single position of the H3K36 (29-43) template sequence was exchanged against 18 natural amino acids (excluding cysteine and tryptophan) to create all possible single amino acid changes of the template sequence. Methylation reactions were performed and the results of each methylation assay were quantitatively analyzed, normalized and the data averaged (Fig. 1a, b). Calculating the standard deviation (SD) of the methylation activity of each single peptide spot indicated a high reproducibility and a good quality of the data, because 90% of the peptides had an SD of less than ±20%, and 60% of the peptides even showed an SD smaller than ±10% (Supplementary Fig. 2). Next, the discrimination factors were calculated for better visualization of the substrate specificity at each position as previously described (Fig. 1c)[34]. The resulting specificity profile shows that NSD2 strongly recognizes the substrate residues from the G33 to P38 of the H3 tail. The results revealed a preference for G at the −3 side, as well as preference for aromatic amino acids and glycine at the −2 position (F > G > Y). At the -1 position, only large aliphatic amino acids (I > L > V) are allowed, whereas at the +1 site few residues are strongly disfavored (A, D, E, G, P). At the +2 position, hydrophobic residues are preferred (V > I > L > P > T).

### Design of an NSD2 specific super-substrate
Surprisingly, we observed at several positions of the NSD2 specificity analysis, that residues different from the natural ones occurring in H3 were preferred, for example K at the −5 site, F at −2, or N at the +3 and +4 sites (Fig. 1a, b). We, therefore, speculated that it might be possible to develop an NSD2 specific H3K36 super-substrate (ssK36), which would provide additional information about the mechanism of NSD2 and its potential novel substrates. To this end, we first selected all strongly methylated single point mutations based on the substrate specificity profile from position −5 to +4 and synthesized the corresponding peptides again on an additional peptide array to validate the observations. As positive control, H3K36 (Fig. 2a; spot A1 and C9) and as negative control, H3K36A (Fig. 2a; spot A2 and C10) were added (Supplementary Table 1). After methylation with NSD2, direct comparison of the methylation signals was performed which reproduced the previous findings in most places. The red circles indicate amino acid exchanges, which led to a strong increase in methylation activity and which were used for the next step of super-substrate development. With an additional array all possible double, triple, quadruple, and quintuple combinations of these activating amino acid changes were further analyzed (Fig. 2b and Supplementary Table 2), again using H3K36 (Fig. 2b; spot A1 and A3) and H3K36A (Fig. 2b; spot A2 and A4) as positive and negative controls. From this array, the sequences of the most strongly methylated

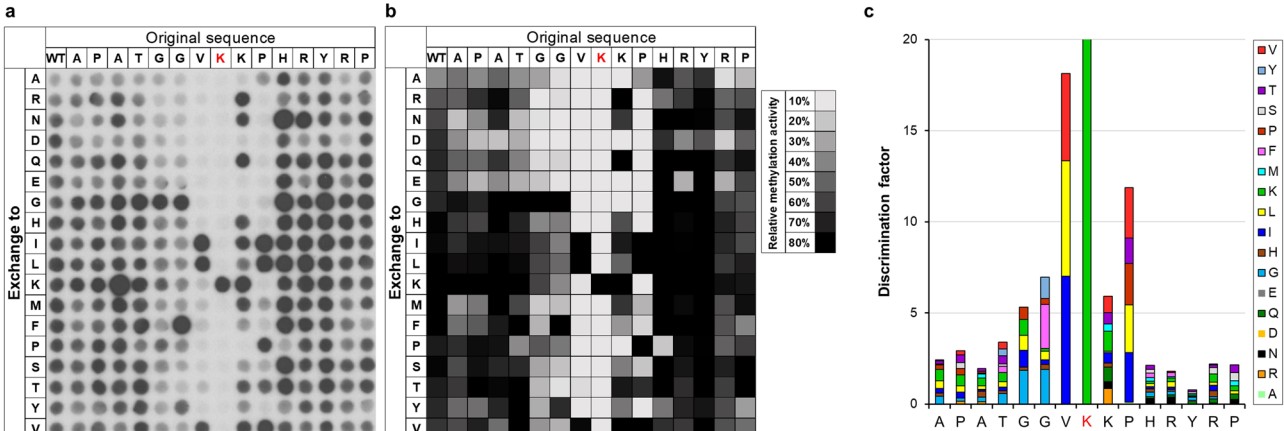

**Fig. 1 | Substrate specificity analysis of NSD2. a** Autoradiographic image (3 days of exposure) of a peptide SPOT array methylated by NSD2 in the presence of radioactively labeled AdoMet. The horizontal axis represents the H3K36 (29–43) template sequence with the target lysine highlighted in red. The vertical axis indicates the residues that were sequentially exchanged at the position corresponding to the row. **b** Data from three independent experiments were averaged after normalizing the full activity to 100%. The activity is displayed in gray-scale as indicated. **c** Discrimination factors for the recognition of each amino acid at the corresponding position of the H3 substrate by NSD2 represented as a bar diagram.

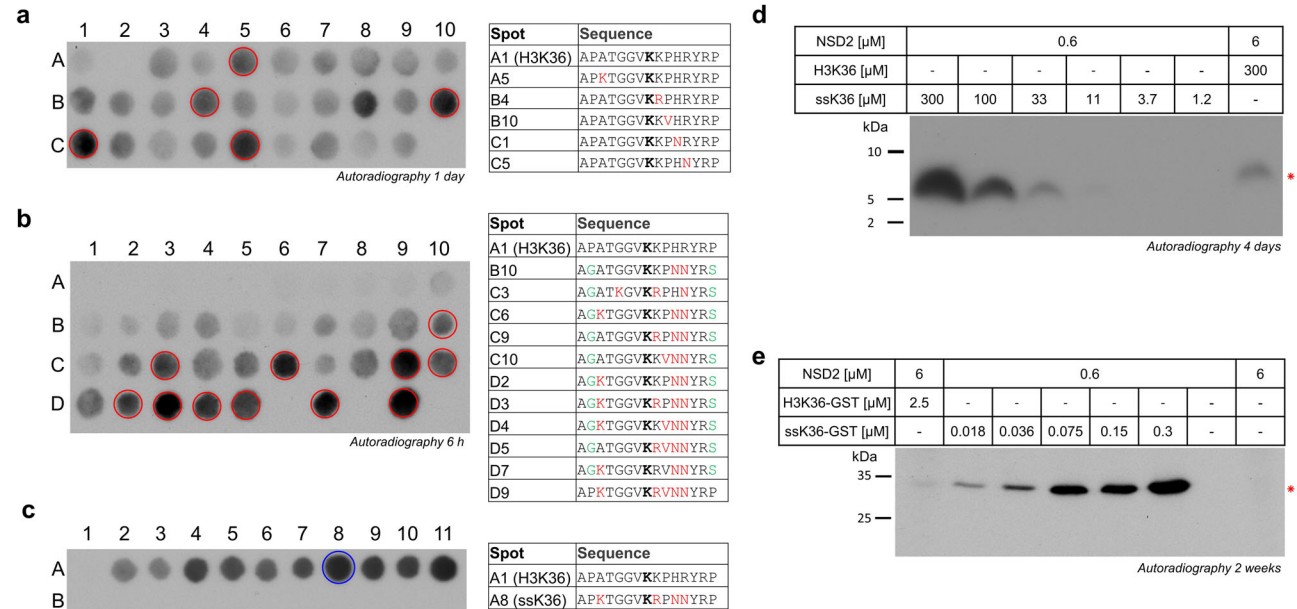

**Fig. 2 | Development of an H3K36 super-substrate specific for NSD2. a** Based on the substrate specificity profile of NSD2 shown in Fig. 1, all strongly methylated sequences with single point mutations from position −5 to +4 were synthesized on an extra array. As positive control, H3K36 (spot A1 and C9) and as negative control, H3K36A (spot A2 and C10) were added. Spots labeled with red circles indicate sequence with single point mutations that were used for the next array in panel (**b**). **b** Peptide array containing all possible double, triple, quadruple, and quintuple amino acid mutation combinations from panel (**a**). As positive and negative controls, H3K36 (spot A1 and A3) and H3K36A (spot A2 and A4) were used. Strongest spots are marked with red circles and further used for the next peptide array shown in panel (**c**). **c** Peptide array containing the peptide sequences selected from panel (b) next to each other in the first line together with their K-to-A mutants in the second line. The best and strongest signal was detected for spot A8 labeled with a blue circle, which is called H3K36 super-substrate (ssK36). It contains 4 amino acid changes when compared with the natural H3 sequence: A31K, K37R, H38N, and R39N. A longer film exposure of the same array also showing the NSD2 activity at the H3K36 peptide is provided in Supplementary Fig. 3a. In panels (**a–c**) the highlighted peptide sequences are shown, all sequences are presented in Supplementary Tables 1–3. **d** Methylation of purified ssK36 and H3K36 peptides. Experiments were performed in 20 μl methylation buffer containing 0.76 μM radioactively labeled AdoMet together with the concentrations of NSD2 and H3K36/ssK36-peptides as indicated. The samples were separated by Tricine-SDS-PAGE and analyzed by autoradiography after 4 days of film exposure. The methylated H3K36 and ssK36 peptide bands are marked with an asterisk. **e** Methylation of ssK36-GST and H3K36-GST proteins. Reactions were conducted in 40 μl methylation buffer containing 0.76 μM radioactively labeled AdoMet together with the concentrations of NSD2 and H3K36-GST/ssK36-GST as indicated. The samples were separated by SDS-PAGE and analyzed by autoradiography after 2 weeks of film exposure. The methylated H3K36-GST and ssK36-GST bands are marked with an asterisk. A protein loading gel was run in parallel using the same protein amounts (Supplementary Fig. 5b).

spots (marked with red circles) were synthesized on a new array in the first line together with the corresponding K-to-A mutant sequences in the second line (Fig. 2c, Supplementary Fig. 3a, and Supplementary Table 3). The methylation data clearly showed that all sequences were much more preferred than the original H3K36 sequence in position A1. Moreover, loss of methylation was observed for all K-to-A mutants indicating that methylation occurred at K36. The best and strongest signal was detected at spot A8 (marked with blue circle). This peptide, which from now on will be

designated as a super-substrate for NSD2 (ssK36), contains 4 amino acid changes when compared with H3K36: K31K, K37R, H38N, and R39N. Further control reactions indicated that ssK36 is also much better methylated than H4K44 and H1.5K168 (Supplementary Fig. 3b). As the H3 peptide contains P at position 2 and at the end, we were concerned that this may cause artifacts. Hence, in some design steps, P30 and P43 were replaced by G and S, but this change did not affect the methylation levels (Supplementary Fig. 4).

Since our lab already designed and investigated an H3K36 super-substrate for the SETD2 PKMT[44,45], we were interested to see if NSD2 is specific for the ssK36 designed here or if it can also methylate the previously developed super-substrate for SETD2. Therefore, a peptide array methylation experiment (Supplementary Fig. 6) was performed, in which the methylation of the NSD2 specific (spot B1) and the SETD2 specific (spot B3) super-substrates, together with their K-to-A mutants (spot B2 and B4, respectively), was directly compared on one array. As in the previous experiments, methylation of the NSD2 super-substrate was much stronger compared to the wildtype H3K36 spotted at positions A1 and A3. Moreover, the methylation data clearly demonstrated that NSD2 only methylates the ssK36 specially designed for NSD2, while no methylation was observed for the SETD2 specific ssK36.

Based on the successful design of an NSD2-specific ssK36 at peptide level, we next intended to investigate its methylation using soluble H3K36 (27-43) and ssK36 (27-43) peptides. Therefore, H3K36 and different dilutions of the ssK36 peptide were incubated with NSD2 in the presence of radioactively labeled AdoMet, separated by a tricine gel and the methylation was analyzed by autoradiography. A strong methylation signal was observed for ssK36 compared to a very weak signal for H3K36 (Fig. 2d). Quantitative analysis of the intensities of the methylated H3K36 and ssK36 bands, considering the substrate amounts and enzyme concentrations, revealed that the ssK36 peptide was methylated about 91-fold more strongly than H3K36. For further comparison of the methylation activities, methylation experiments were performed at protein level. For this, GST-tagged H3K36 (29-43) and ssK36 (29-43) were cloned, overexpressed and purified. For the protein methylation assay, H3K36-GST and ssK36-GST were incubated with NSD2 in the presence of radioactively labeled AdoMet, separated by SDS-PAGE, and the methylation was analyzed by autoradiography. As shown in Supplementary Fig. 5a, a strong methylation signal of the GST-tagged ssK36 was detected, but no signal for the wildtype H3K36 protein. To determine the increase of ssK36-GST methylation as compared to H3K36-GST quantitatively, additional methylation experiments were conducted using stepwise reduced amounts of ssK36-GST (Fig. 2e). A protein loading

gel was run in parallel using the same protein amounts (Supplementary Fig. 5b). Quantitative analysis of the intensities of the methylated H3K36-GST and ssK36-GST bands considering the substrate amounts and enzyme concentrations revealed that ssK36-GST was methylated about 3000-fold more strongly than H3K36-GST. This confirms that NSD2 strongly prefers the NSD2 super-substrate for methylation also at protein level.

## MD simulations of NSD2-peptide complexes

Methylation experiments showed that NSD2 has a >100-fold higher activity towards the artificially designed super-substrate peptide ssK36 compared to the canonical H3K36 peptide. We next applied molecular dynamics (MD) simulations to reveal the molecular mechanism behind this massive increase in NSD2 activity. For this, we used the structure of NSD2 complexed with the H3 histone tail from PDB 7CRO as the starting point. The peptide in this complex was elongated using PDB 5V21 as a structural template. By mutating the H3K36 peptide sequence, the NSD2-ssK36 complex was modeled. The target K36 was manually deprotonated as required for the $S_N2$ mechanism that is used for the methyl group transfer[43]. NSD2 in complex with ssK36 and H3K36 was subjected to MD simulations (50 replicates of 100 ns each) and frames were recorded every 20 ps (Fig. 3a). In order to define criteria describing the likelihood of a methyl group transfer during the MD simulations and, by this, approximate the enzymatic NSD2 activity, the geometric requirements for a transition state (TS)-like conformation were applied, which were derived from the known $S_N2$ geometry of methyl group transfer reactions (Supplementary Fig. 7a)[32,43,45]. Strikingly, the complex of NSD2-ssK36 established significantly more TS-like conformations than the complex of NSD2 with H3K36 (Fig. 3b). The difference between the two peptides was even more pronounced after sorting the simulation replicates into bins. In 36 out of 50 replicates, the ssK36 peptides established >1000 TS-like conformation frames (within the 5000 frames in total per replicate) indicating that it had adopted a very active conformation in which the TS-like state was reached frequently. This state was observed only once in the case of H3K36 (Fig. 3c). Conversely, the majority (41 out of 50 replicates) of H3K36 simulations had 0-250 TS-like conformation frames, whereas only 4 of the ssK36 simulation replicates had such a low number of TS-like conformation frames. These observations indicate that the 4 amino acid changes in ssK36 cause a much better stabilization of TS-like conformations of the target lysine together with AdoMet because it can adopt a hyperactive conformation. Comparison of exemplary snapshots of complex structures revealed that the ssK36 peptide tends to bend towards the active site of NSD2 and the bound AdoMet, a structural change that was not observed with the H3K36 peptide (Fig. 3a). A movie of an MD

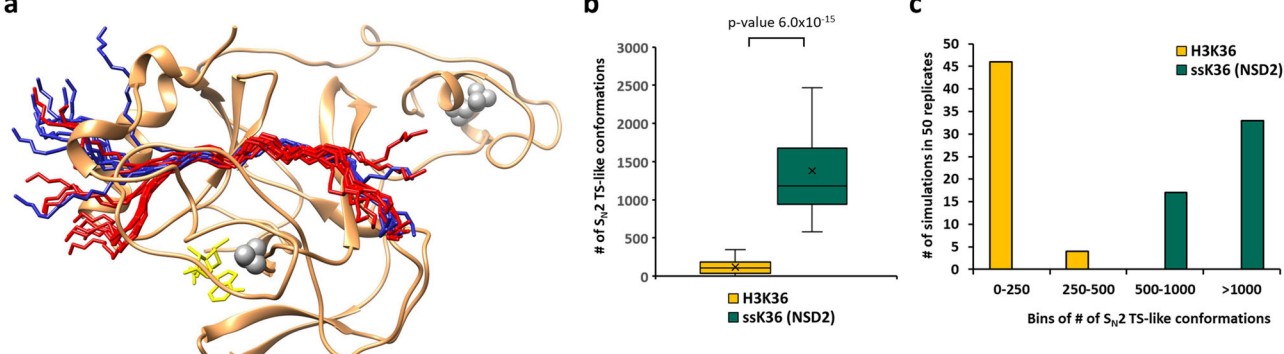

**Fig. 3 | The NSD2-ssK36 peptide complex establishes more $S_N2$ TS-like conformations than the NSD2-H3K36 complex. a** Superposition of 10 randomly selected peptide structures taken from the ssK36 and H3K36 MD simulations after superposition of the NSD2 proteins. The peptides are shown in red (ssK36) and blue (H3K36) ribbon. For clarity, only one NSD2 structure is shown in tan ribbon, with Zinc ions in gray and AdoMet in yellow. **b** Average number of $S_N2$ TS-like conformations observed in the 50 MD simulation replicates of 100 ns for each peptide

complexed to NSD2 and AdoMet. Boxes show the median, 1st and 3rd quartile ($n = 50$ independent MD simulations). Whiskers display the 1.5 IQR distance. The p-value was determined by a two-sided T-test with unequal variance. **c** Histogram of simulation replicates shown in panel (**b**), where every replicate was sorted into bins depending on how many TS-like conformations were observed during the simulation run.

simulation replicate of NSD2 complexed with the ssK36 peptide and AdoMet is also provided on DaRUS (https://doi.org/10.18419/darus-3815).

As described above, in these MD simulation experiments 50 replicates were conducted for each system (NSD2-H3K36 and NSD2-ssK36) after multiple equilibration steps of each system before production runs. However, production runs were started from the same equilibrated state of each system. To ensure, that the initial state after equilibration did not influence the outcome of the production runs, 10 additional replicates for each system were conducted. Here, each replicate had an individual equilibration (Supplementary Fig. 7c). The results of these additional 10 replicates with individual equilibration match the results of the 50 production runs (Supplementary Fig. 7d). Finally, MD simulations of peptides containing the P30G and P43S mutations as used for some of the peptide methylation experiments (P30G, P43S double mutant) were conducted as well and the simulation results available at DaRUS (https://doi.org/10.18419/darus-3815) also match the previously described data.

## Contact analysis of H3K36- and ssK36-NSD2 complexes
To investigate the molecular mechanisms behind the better stabilization of the TS-like conformations in the ssK36-NSD2 complex, a contact map of the established contacts between peptide and protein during the simulation was prepared using contact map explorer[46]. The analysis was based on distance criteria, and contacts were considered if the distance of a pair of heavy atoms from the peptide and an NSD2 residue was below 4.5 Å. The fraction of time in which a contact was established was measured and a contact profile created. The resulting contact maps for H3K36 and ssK36 were contrasted and differences extracted (Fig. 4).

The largest deviations between the two peptides were found at three of the four mutated residues. The larger side chain of R37 in ssK36 (when compared to K37 in H3K36) enables contacts primarily with the E1216 side chain and the K1220 and K1221 backbone atoms (Fig. 5a). H39 in H3K36 is positioned in a pocket surrounded by NSD2 residues P1146-T1150, while ssK36-N39 points outwards into to solvent leading to the loss of this interaction (Fig. 5b). Moreover, a higher contact frequency of ssK36-P38 with NSD2 residues F1177-Y1179 next to the H39 pocket is observed. P38 in H3K36 is oriented differently than in ssK36, potentially influencing the structure of the backbone atoms in the peptide. Hence, for K31, R37, and P38 distinct contacts are established in ssK36, potentially stabilizing the TS-like conformation. Conversely, our data suggest that the H39 contact to NSD2 residues P1146-T1150 is unfavorable for TS-like conformations and loss of this interaction by the change of H3K36-H39 to ssK36-N39 stimulates catalytic activity. Finally, the A31K mutation in ssK36 causes the peptide residues P30-T32 to strongly interact with NSD2 residues H1110-F1117 and E1187-T1189. The K31 side chain amino group is solvent exposed and interacts with the T1116 hydroxyl group (Fig. 5c). This interaction likely triggers a bending of the N-terminal part of the ssK36 peptide (A29-G33) towards the active site and AdoMet. In contrast, this part of the H3K36 peptide makes contacts with I1106 and L1184-N1186, which keeps the peptide in a straight orientation.

As a consequence of these altered interactions, the mutated residues change the overall conformation of the ssK36 peptide in the NSD2 binding cleft. Especially at the N-terminus, ssK36 bends towards the AdoMet, whereas H3K36 stays straight (Figs. 3a and 5c). Due to this conformational change, ssK36-K31 can also interact with the hydroxyl group of the AdoMet sugar moiety. Moreover, the SET-I loop (D1114–Y1119), which is suspected to contribute to the substrate specificity of PKMTs[47,48], changes its conformation towards AdoMet and H1116 interacts with the AdoMet sugar moiety as well (Fig. 5c). These interactions could stabilize AdoMet in the cofactor binding pocket, bridge between AdoMet and the bound ssK36 peptide and hence increase the probability of a methyl group transfer. This hypothesis was further validated by the finding that the fluctuations (RMSF) of the AdoMet heteroatoms during the MD simulations are strongly reduced in the NSD2-ssK36 complex, as compared to NSD2-H3K36 (Supplementary Fig. 7b). This observation suggests that the AdoMet is stabilized in a more catalytically competent conformation in the NSD2-ssK36 complex by the additional contacts which are specific for the ssK36 complex.

## Identification and methylation of novel non-histone peptide substrates of NSD2
Next, we were interested to use the specificity profile of NSD2 to identify novel substrates of NSD2. To this end, a ScanSite search[49] with the specificity sequence motif was performed to identify human proteins that contain a sequence matching the NSD2 sequence specificity profile. In order to cover most possible substrates, the search profile was expanded at this stage, also including less preferred but still allowed residues at several positions (Table 1).

Since NSD2 is mainly localized in the nucleus (www.proteinatlas.org/ENSG00000109685-NSD2, version 23.0), the search was restricted to nuclear proteins, which led to the identification of 226 substrate candidates. A peptide array with 15 aa long peptides of each candidate with the target lysine centered in the middle was synthesized using the SPOT synthesis method (Fig. 6a, Supplementary Data 1). As positive controls, H3K36 (aa 29-43) (spot A1) and H4K44 (spot A3) were included and the corresponding K-to-A mutations served as negative controls (spot A2 and A4, respectively). The peptide array was incubated with NSD2 using radioactively labeled AdoMet and the methyl transfer was detected by autoradiography revealing numerous methylation signals. Depending on the strength of the methylation signal and the biological relevance of the target protein, 25 peptides were selected and further analyzed by additional peptide arrays to determine if the predicted target lysine is the site of modification. For this, pairs of the 15 aa long WT sequences and the respective K-to-A mutants were prepared and incubated with NSD2 in the presence of radioactively labeled AdoMet (Fig. 6b, Supplementary Table 4). Methylation was validated in all of the peptides, and activity at the target lysine was confirmed in all but 3 of the candidate substrates which showed no loss of methylation in the K-to-A mutant indicating that the methylation did not occur at the predicted target lysine in these peptides. All three peptides contain additional lysine residues, which may be methylated by NSD2. To confirm this and identify the true target residue, additional experiments would be required. In summary, 22 peptide substrates of NSD2 with methylation at the predicted target lysin were discovered, whose methylation by NSD2 was not known before to the best of our knowledge.

## Investigation of the methylation of non-histone protein substrates by NSD2
It is possible that some of the methylated peptide substrates are not methylated at protein level, because the NSD2 target site is rich in hydrophobic residues which might be buried in folded proteins making them inaccessible. For this reason, we cloned the 22 non-histone substrate candidates proteins identified above with a GST-tag, expressed them in E. coli, and finally 17 of them were successfully purified. Roughly comparable amounts of the purified proteins were verified by SDS-PAGE and Western Blot using an anti-GST antibody (Supplementary Fig. 8a, b) and used for subsequent protein methylation assays. After incubation of the substrate proteins with NSD2 and radioactively labeled AdoMet, the samples were separated by SDS-PAGE and methyl transfer was detected by autoradiography (Supplementary Fig. 8c). For two of the targets, ATRX and FANCM, a methylation signal was detected and loss of methylation of the corresponding K-to-R mutants (Fig. 6c) validated the methylation at the target lysine residues K1033 of ATRX and K819 of FANCM, which were identified in the peptide methylation experiments. Additional methylation experiments revealed that methylation of ATRX was comparable to that of H3K36, while methylation of FANCM was weaker (Supplementary Fig. 9).

We have inspected structures of AlphaFold models[50] of the substrate proteins to determine if the target lysine is placed in a secondary structure element or in an unfolded part of the proteins (Supplementary Table 5). This analysis revealed that ATRX and FANCM were the only proteins among our candidates where the target lysine resides in a region shown or predicted to be unfolded.

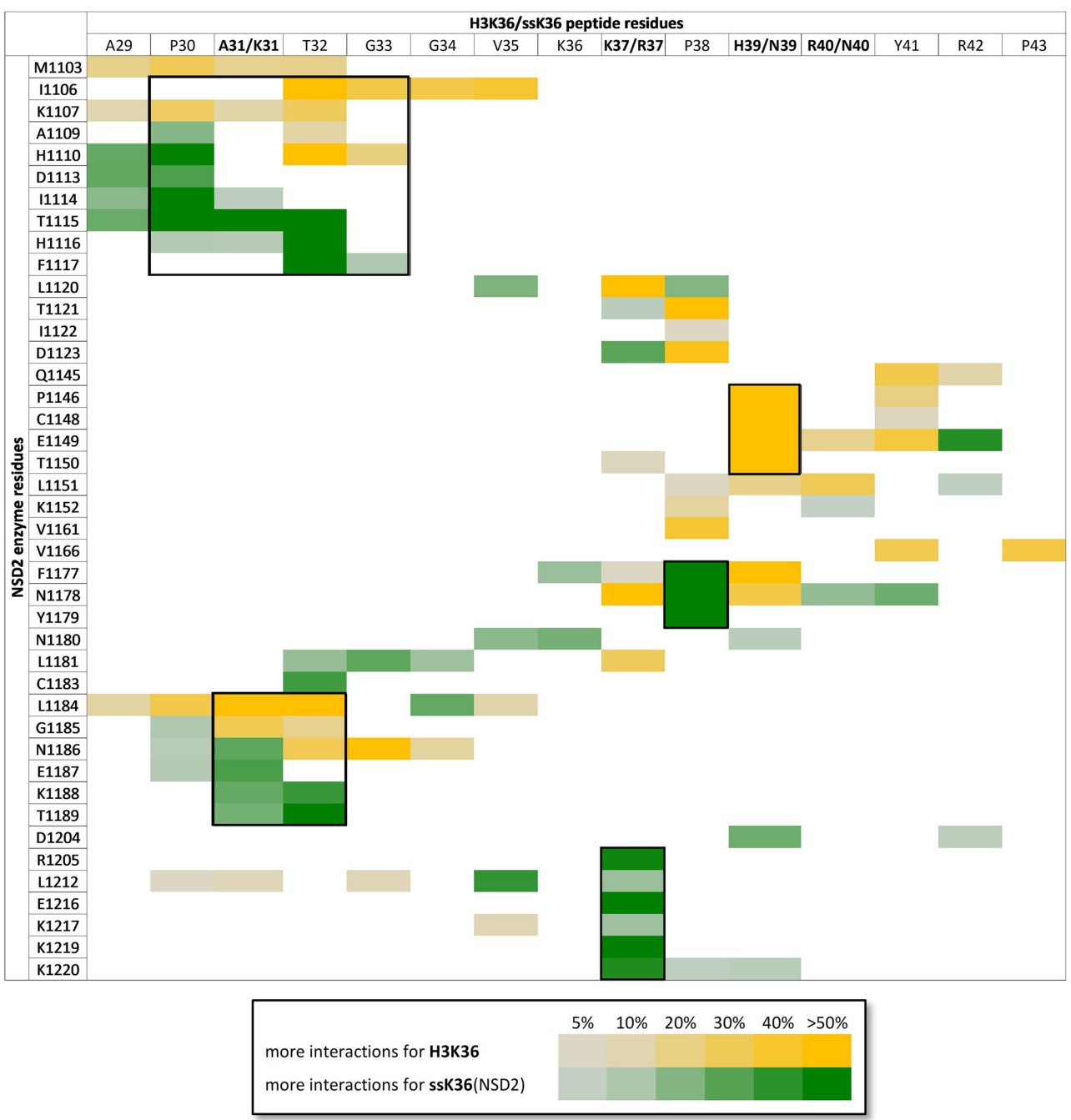

**Fig. 4 | The H3K36 and ssK36 peptides establish different contacts with NSD2.** The figure shows the contact map difference as the result of the subtraction of the H3K36 and ssK36 contact map including all NSD2 residues which exhibited noticeable changes. Yellow indicates that a specific contact was more often observed in simulations with H3K36. Green symbolizes a higher contact frequency for ssK36. Framed regions highlight the largest difference.

To analyze methylation of ATRX and FANCM by NSD2 at the cellular level, we aimed to apply an H3K36me1 antibody, because the methylation sites in ATRX and FANCM have a similar sequence context as H3K36. To investigate if the anti-H3K36me1 antibody was also able to detect the NSD2 methylation of ATRX and FANCM, in vitro protein methylation reactions were performed by incubating either ATRX or FANCM with or without NSD2 enzyme in methylation buffer containing unlabeled AdoMet. The samples were separated by SDS-PAGE, transferred to a nitrocellulose membrane and the membrane was probed with the anti-H3K36me1 antibody (Supplementary Fig. 10). A clear signal was observed for the methylated FANCM protein, only when NSD2 was present. For ATRX, the H3K36me1 antibody was not able to discriminate perfectly between

methylated and unmethylated proteins, but the signal for the methylated ATRX protein was much stronger than for the unmethylated protein. Additionally, the automethylated NSD2 was stained by the antibody, demonstrating that it has a rather broad Kme1 reactivity.

In the next step, ATRX and FANCM were cloned into the mammalian expression vector pEYFP-C1 and the full-length NSD2 enzyme into the mammalian expression vector pECFP-C1. Both plasmids were individually transfected into HEK293 cell and the expression of all proteins was validated by Western blot using antibodies against the fluorophore tags (Supplementary Fig. 11). For methylation analysis, HEK293 cells were co-transfected with two plasmids encoding CFP-tagged NSD2 enzyme and YFP-tagged ATRX or FANCM. As a control, HEK293 cells were only

**Fig. 5 | Example structures of the 15 aa long H3K36 (yellow) and ssK36 (green) peptides in the NSD2 (gray) peptide binding cleft taken from the MD simulations.** In the overview image, NSD2 is shown in the starting conformation. In the panels (**a**–**c**) example structures illustrate the global differences in the contact maps of the NSD2-H3K36 and NSD2-ssK36 complexes as shown in Fig. 4. The target lysine (pink) is inserted into the hydrophobic tunnel of the NSD2 SET domain. AdoMet (orange) binds from the opposing site. **a** H39 of H3K36 is positioned in a pocket made by P1146, C1148, E1149, and T1150. In contrast, ssK36-N39 points into the solvent which positions P38 closer towards residues F1177, N1178, and Y1179. **b** The longer side chain of R37 in ssK36 contacts E1216, which is not possible for K37 in H3K36. **c** K31 in ssK36 interacts with NSD2 T1116, whereas residues T32-V35 in H3K36 interact with I1106.

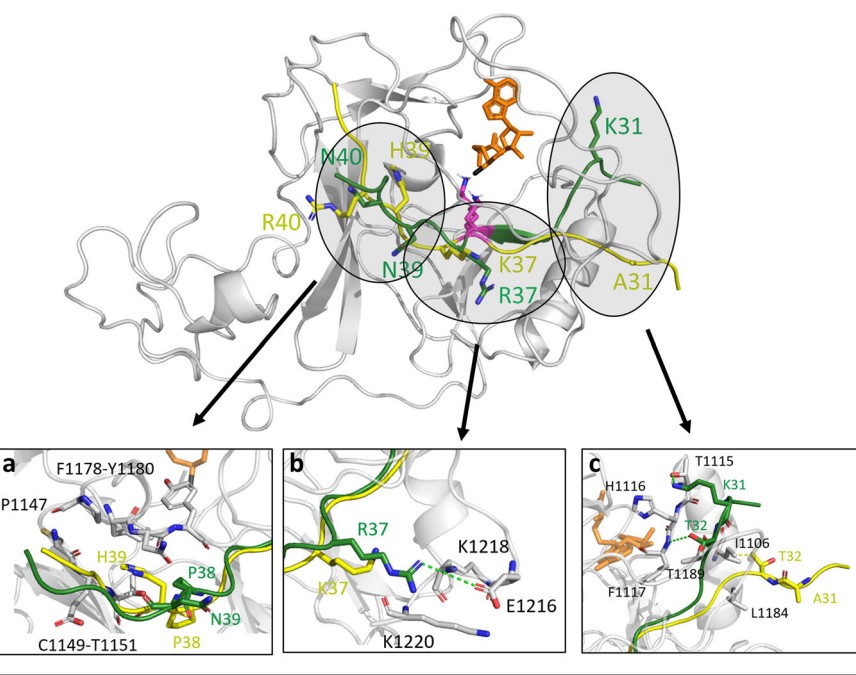

## Table 1 | Motif used for the search for novel NSD2 non-histone substrates

| Site | −1 | K | +1 | +2 | +3 | +4 |
|---|---|---|---|---|---|---|
| Motif | ILV | K | KRVQNI | VILP | NGLSFTMIHQAEK | LNQGHIKMF |

transfected with one of the substrate protein expression plasmids without the NSD2 plasmid. After isolation of the YFP-tagged substrate proteins by GFP trap, equal loading of methylated and unmethylated sample was analyzed by Western Blot using an anti-GFP antibody (Fig. 6d). The cellular methylation of ATRX and FANCM was investigated using the previously validated H3K36me1 antibody. The results showed methylation of ATRX and FANCM, which were isolated from cells cotransfected with NSD2, whereas no methylation signal was observed without NSD2 cotransfection. This result indicates that NSD2 is the responsible methyltransferase for ATRX and FANCM methylation in HEK293 cells establishing both proteins as novel non-histone substrates of NSD2 whose methylation by NSD2 has not been demonstrated before to the best of our knowledge.

## Discussion

Investigation of the substrate specificity of PKMTs has important implications by providing details about the mechanism of peptide interaction and catalysis and allowing for unbiased searchers for novel PKMT substrates[43]. In this work, we have investigated the substrate specificity of the NSD2 PKMT, which is a well-established H3K36 mono- and dimethyltransferase with important roles in chromatin regulation, cell physiology, and cancer. Our data revealed a highly specific interaction of NSD2 with the amino acid residues between the −3 and +2 sites surrounding the K36 target lysine. Interestingly, we observed at several positions of the H3K36 sequence that amino acid residues differing from the natural ones in the H3K36 context were preferred by NSD2. A combination of four of these preferred residues led to the design of a super-substrate (ssK36) that was methylated about 100 times more strongly at peptide level and even more preferred as GST-tagged substrate protein. MD simulations revealed that the super-substrate peptide adopts a bent conformation in complex with NSD2 that may help K36 to approach the AdoMet more efficiently. This conformational change was driven by distinct contacts established between the ssK36 and NSD2 which support catalysis and loss of unfavorable contacts engaged by H3K36.

Moreover, AdoMet was stabilized in a catalytically competent conformation by specific contacts only observed in the NSD2-ssK36 complex MD simulations. The increased preference for ssK36 at protein level could be connected to the possibility that the substrate peptide part can dock on the protein body which makes access for NSD2 more difficult thereby enhancing the positive effects of ssK36.

It is interesting to consider that the natural H3K36 methylation by NSD2 takes place in the nucleosomal context, where the histone tail binding occurs together with interactions of the enzyme with the other parts of the nucleosome and the linker DNA. In this reaction, the H3 tail must be partially lifted up in order to make K36 accessible for methylation[31,51]. This may require special conformational adaptations that are not ideal for the interaction with the isolated peptide, explaining why alternative peptide sequences can be better substrates. This mechanism could explain, why a super-substrate could also be designed for SETD2, another H3K36 methyltransferase[32,44].

We have identified 22 novel NSD2 substrate candidates that contain a lysine residue in a sequence context favored by NSD2 but were not known to be methylated by NSD2 before to the best of our knowledge. All of them were strongly methylated at the peptide level, still only two were methylated at the protein level in vitro. Comparison of the structures of these proteins revealed that ATRX and FANCM were the only proteins containing the target lysine in a region shown or predicted to be unfolded. This finding suggests that NSD2 cannot access target peptides that are part of a secondary structure. This observation can be explained, because the hydrophobic residues at the −1 and +2 sites of the NSD2 specificity profile are likely to be embedded in the hydrophobic core of the folded protein and not accessible for NSD2 interaction.

The role of NSD2 in the Wolf-Hirschhorn syndrome and H3K36 methylation implicated a direct connection to DNA repair. This conjecture was further validated when it was discovered that the interactome of NSD2 comprises many factors involved in the DNA repair including Poly(ADP-ribose) polymerase 1 (PARP1)[52]. Subsequently, it was found that PARylation reduces NSD2 histone methyltransferase activity and impedes its chromatin binding[52]. Moreover, different previously identified non-histone substrates of NSD2 have important roles in DNA damage repair such as PTEN[27] and Aurora kinase A (AURKA)[26], and other studies showed that NSD2 enhances DNA damage repair leading to an increase in resistance to chemotherapeutic agents[53]. This connection between NSD2 and DNA

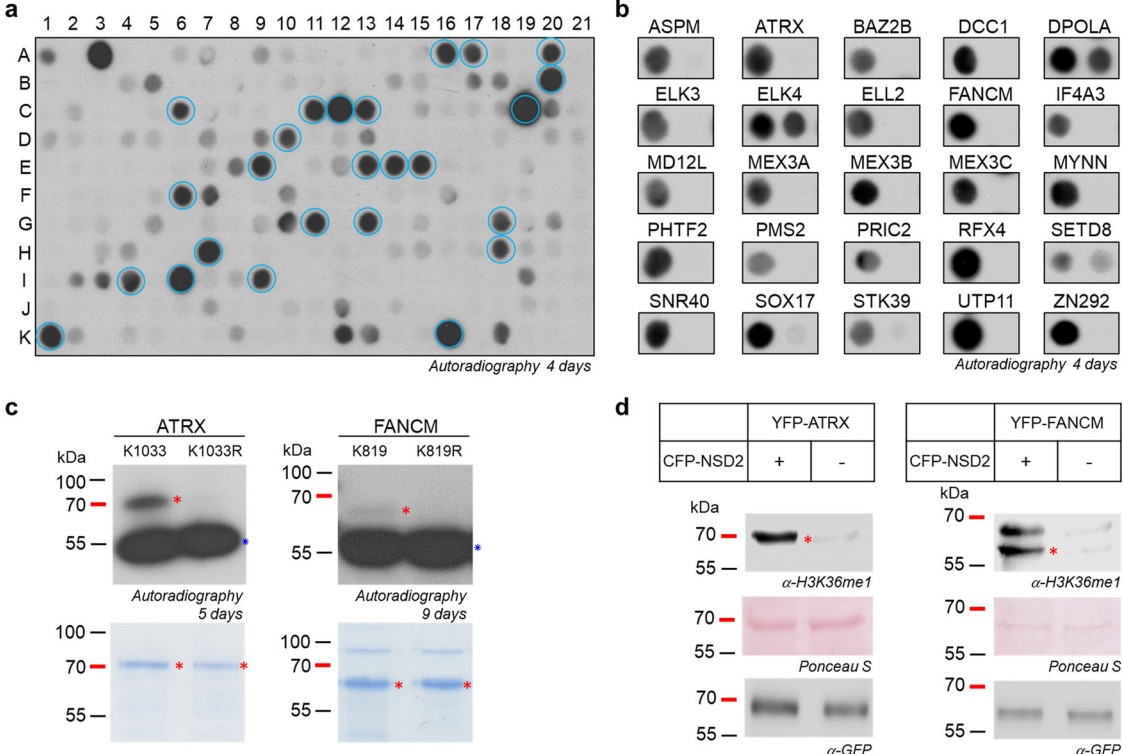

**Fig. 6 | Identification of NSD2 non-histone substrates. a** 15 aa long peptides (with the target K in the middle) of potential non-histone substrates identified in ScanSite searches[49] were synthesized on a SPOT peptide array and methylated by NSD2. As positive controls, H3K36 (aa 29–43) and H4K44 (aa 161–175) were included in spot A1 and A3. As negative controls, the corresponding K-to-A mutants were used: H3K36A (aa 29–43) in spot A2 and H4K44A (aa 161–175) in spot A4 (Supplementary Data 1). 25 substrate peptides were selected for further analysis (marked with blue circles) based on their methylation strength and the biological relevance of the corresponding proteins. **b** To determine the correct target site position additional peptide array methylation experiments were performed, including the K-to-A mutants of each selected candidate substrate (Supplementary Table 4). Methylation of the predicted target lysine was validated in all but 3 of the candidate substrates (DPOLA, ELK4 and SETD8) which showed no loss of methylation in the K-to-A

mutant. **c** Purified wildtype and K-to-A mutant proteins of ATRX and FANCM were methylated by NSD2. Equal substrate protein amounts were verified by Coomassie staining and the bands of expected sizes are marked with red asterisk. The autoradiographic image after 5 days for ATRX and 9 days for FANCM shows methylation in the wildtype protein and loss of methylation in the corresponding mutants. Automethylation of NSD2 is labeled with a blue asterisk. NSD2 concentration was 2.5 µM, ATRX and FANCM concentrations were 6 and 3 µM. **d** HEK293 cells were transfected with YFP-tagged ATRX or FANCM with or without CFP-tagged NSD2. After cell lysis, the substrate proteins were purified by GFP trap and equal loading of the sample was verified by Ponceau S staining and Western Blot using anti-GFP antibody. To determine ATRX and FANCM methylation at cellular level, Western Blot with the previously verified anti-H3K36me1 antibody (Supplementary Fig. 10) was performed.

damage repair is further enhanced by our finding that ATRX and FANCM are direct targets of NSD2, because these two proteins are helicases with important roles in DNA repair and R-loop metabolism[54].

The specificity analysis of NSD2 was based on the methylation of peptides containing all single mutations of the H3K36 template sequence. The elevated peptide methylation levels of the H1.5K168 peptide (when compared with H3K36) cannot be explained on the basis of this analysis, suggesting that combined effects of two or more amino acid changes in the target sequence affect NSD2 activity. This could be investigated in methylation studies of peptide arrays containing combinatorial mutations in future. So far, super-substrates have only been designed for PKMTs acting on H3K36 and more systematic studies will be required to find out, if super-substrate can also be designed for PKMTs that methylate other primary targets. In vitro and in vivo methylation studies were conducted with tagged proteins or protein domains in this work. Future studies should aim to investigate methylation of the full-length endogenous proteins. Due to the limited activity of NSD2, all in vitro kinetics were conducted under single-turnover conditions precluding the measurement of $K_m$-values of peptide and AdoMet binding. We observed that only protein substrate candidates bearing the target lysine in unfolded regions were methylated by NSD2 in vitro. However, this does not rule out methylation of other substrates in cells, either during the protein folding process or in the presence of chaperones, which can help NSD2 to get

access to its target region. Future research needs to address this question with cellular methylation studies. We discovered that the helicases ATRX and FANCM which have important roles in DNA repair and R-loop metabolism are direct methylation targets of NSD2. Future work will be needed to unravel the biological effects of NSD2 methylation on these factors on DNA repair.

## Methods

### Cloning, expression, and purification of proteins

The DNA sequences encoding for the human NSD2 enzyme (aa 991–1240; Swiss Prot No. O96028) and the putative human substrate protein domains were amplified by PCR using cDNA isolated from HEK293 cells. Protein domains of the non-histone substrates were designed with the Scooby domain prediction tool (http://www.ibi.vu.nl/programs/scoobywww/)[55]. All constructs were cloned into the pGex-6p-2 expression vector as GST-fusion proteins. The K-to-R mutations of the non-histone substrates were introduced using a megaprimer PCR mutagenesis method[56]. The ssK36 (29-43)-GST construct was cloned by side-directed mutagenesis using the H3K36 (29-43)-GST plasmid as template taken from ref. 44. For mammalian expression, the coding sequence of the full-length NSD2 (Swiss Prot No. O96028) was cloned into the pECFP-C1 (Clontech, USA). The protein domains encoding for ATRX (aa 893–1188; Swiss Prot No. P46100) and FANCM (aa 723–933; Swiss Prot No. Q8IYD8) were cloned into the

pEYFP-C1 vector (Clontech, USA). All cloning steps were confirmed by Sanger sequencing.

For protein overexpression, the plasmids were transformed into *E. coli* BL21-CodonPlus (DE3) cells (Novagen, USA) which were grown in LB medium at 37 °C until an $OD_{600}$ of 0.6 to 0.8 was reached. The culture was then shifted to 20 °C overnight (14–16 h) and protein expression was induced with 1 mM isopropyl-D-thiogalactopyranoside (IPTG). Afterward, the cells were harvested by centrifugation at 4,500 rpm, washed once with STE buffer (10 mM Tris-HCl pH 8, 1 mM EDTA and 100 mM NaCl) and the cell pellet was stored at −20 °C. For purification, the cell pellet was thawed on ice, resuspended in sonication buffer (50 mM Tris/HCl pH 7, 150 mM NaCl, 1 mM DTT, 5% (w/v) glycerol) supplemented with protease inhibitor cocktail and lysed by sonication (14 rounds, 30% power, 4 °C). Thereafter, the sample was centrifuged at 18,000 rpm for 90 min and the supernatant was loaded onto a Glutathione Sepharose 4B resin (GE Healthcare) column, which was pre-equilibrated with sonication buffer. Afterward, the beads were washed once with sonication buffer and twice with washing buffer (50 mM Tris/HCl pH 8, 500 mM NaCl, 1 mM DTT, 5% (w/v) glycerol). Subsequently, the bound proteins were eluted with elution buffer (40 mM reduced glutathione, 50 mM Tris/HCl pH 8, 500 mM NaCl, 1 mM DTT, 5% (w/v) glycerol) and then dialyzed against low glycerol dialysis buffer 1 (20 mM Tris/HCl pH 7.4, 100 mM KCl, 0.5 mM DTT, 10% (w/v) glycerol) for 3 h and afterwards overnight against high glycerol dialysis buffer 2 (20 mM Tris/HCl pH 7.4, 100 mM KCl, 0,5 mM DTT, 60% (w/v) glycerol). The purified proteins were analyzed by sodium-dodecyl-sulfate-polyacrylamide gel electrophoresis (SDS-PAGE).

### Peptide array synthesis
Peptide arrays containing 15 aa long peptides were synthesized on cellulose membrane using the SPOT synthesis method[57] with an Autospot Multipep synthesizer (Intavis AG). Each spot contained ~9 nmol peptide (Autospot Reference Handbook, Intavis AG), and the successful synthesis of the peptides on the cellulose membrane was qualitatively confirmed by bromophenol blue staining[33,34].

### Peptide array methylation
All peptide arrays were preincubated in methylation buffer containing 50 mM Tris/HCl pH 8.5, 50 mM NaCl, and 0.5 mM DTT for 5 min at room temperature. Then, the membranes were incubated in methylation buffer supplemented with 0.4 µM NSD2 and 0.76 µM labeled [methyl-$^3$H]-AdoMet (Perkin Elmer Inc., dissolved at 25 µM in 10 mM sulfuric acid) for 60 min at 25 °C on a shaker. Afterwards, the arrays were washed five times with 100 mM NH₄HCO₃ and 1% SDS, followed by incubation for 5 min in Amplify NAMP100V solution (GE Healthcare). The membranes were exposed to Hyperfilm™ high performance autoradiography films (GE Healthcare) in the dark at −80 °C. Film development was performed with an Optimus TR developing machine. Spot intensities were determined by ImageJ[58] and normalized between 0 and 100% for the highest and lowest values of each membrane. Data from independent arrays were averaged and the SD calculated for each spot using Excel. Discrimination factors were calculated as described previously[33,34]. The discrimination factor D(x,i) describes the relative preference of NSD2 for each amino acid i at position x of the recognition sequence (Eq. 1):

$$D(x, i) = (v(i)/\langle v(j \neq i) \rangle) - 1 \qquad (1)$$

where $v$(i) is the rate of methylation of the peptide variant carrying amino acid i at position x, and $\langle v(j \neq i) \rangle$ is the average rate of methylation of all other peptides carrying a different amino acid at position x (including the wild-type sequence). The D values directly allow the recognition of each residue in the substrate peptide to be compared and displayed quantitatively.

### Peptide methylation assay
For peptide methylation experiments, the H3K36 (27-43: Ac-APATGGVK KPHRYRP-NH2) and the ssK36 (27-43: Ac-AGKTGGVKRPNNYRS-NH2) peptides were purchased from Royobiotech (Shanghai, China) in HPLC grade purified form with >95% purity. 6 µM NSD2 were used for the methylation of 300 µM H3K36 peptide substrate and 10 times less NSD2 enzyme for the methylation of the ssK36 peptide used in a concentration ranging from 1.2 to 300 µM in 20 µl methylation buffer (50 mM Tris/HCl pH 8.5, 50 mM NaCl and 0.5 mM DTT) supplemented with 0.76 µM radioactive labeled AdoMet (Perkin Elmer). Reactions were conducted overnight at 25 °C. The reactions were stopped by the addition of 2X Tricine-SDS-PAGE loading buffer and incubation for 5 min at 95 °C. Afterward, the samples were separated by Tricine-SDS-PAGE, which was followed by the incubation of the gel in amplify NAMP100V (GE Healthcare) for 1 h on a shaker and drying of the gel for 2 h at 70 °C in vacuum. The signals of the transferred radioactive labeled methyl groups were detected by autoradiography using a Hyperfilm™ high performance autoradiography film (GE Healthcare) at −80 °C in the dark. Film development was performed with an Optimus TR developing machine. Band intensities were quantified by ImageJ[58] and background corrected.

### Protein methylation assay
The methylation of non-histone substrate proteins and recombinant H3.1 (NEB) was performed in methylation buffer (50 mM Tris/HCl pH 8.5, 50 mM NaCl and 0.5 mM DTT) supplemented with NSD2 and 0.76 µM labeled [methyl-$^3$H]-AdoMet (Perkin Elmer Inc., dissolved at 25 µM in 10 mM sulfuric acid) for 4 h at 25 °C. For methylation of the H3K36 (29-43)-GST and ssK36 (29-43)-GST proteins, 420 nM NSD2 was added and the mixture incubated overnight at 25 °C. The methylation reaction was stopped by the addition of SDS loading buffer and boiling for 5 min at 95 °C. NSD2 and target protein concentrations in the individual experiments are indicated in the figure legends. Equal amounts of target protein and K-to-R mutant protein was confirmed by Coomassie Brilliant Blue staining and Western Blot using as primary antibody anti-GST (GE Healthcare, 27457701 V). The methylated samples were separated by 12% SDS-PAGE. Then, the SDS gel was incubated for 1 h in Amplify NAMP100V (Ge Healthcare) and dried for 90 min at 65 °C under vacuum. The dried SDS gel was exposed to Hyperfilm™ high performance autoradiography films (GE Healthcare) in the dark at −80 °C. Film development was performed with an Optimus TR developing machine. Band intensities were quantified by ImageJ[58] and background corrected.

### Cell culture, transfection and immunoprecipitation
HEK293 cells were obtained from German Collection of Microorganisms and Cell Cultures GmbH (https://www.dsmz.de/). They were grown in Dulbecco's Modified Eagle's Medium (Sigma) supplemented with 5% fetal bovine serum, penicillin/streptomycin, and L-glutamine (Sigma) in an incubator providing 37 °C and 5% CO₂. The pECFP-C1 tagged full-length NSD2 was co-transfected with pEYFP-C1 fused ATRX or FANCM using polyethylenimine (Polyscience, USA; according to manufacturer´s instructions). 72 h after transfection, the cells were washed with PBS buffer and harvested by centrifugation at 500 g for 5 min. For methylation analysis, the YFP-fused ATRX and FANCM substrate proteins were immunoprecipitated from cell extract using GFP-Trap® A (Chromotek) following the manufacturer´s instructions. The samples were heated to 95 °C for 5 min in SDS-gel loading buffer and resolved by 16% SDS-PAGE. Analysis was performed by Western Blot using as primary antibody H3K36me1 (Abcam, UK; Cat. No: ab9048) or GFP antibody (Clontech, lot. 1404005).

### MD Simulation of the NSD2-peptide complexes and trajectory analysis
All molecular dynamics (MD) simulations were performed in OpenMM 7.5.1[59,60] utilizing the NVIDIA CUDA[61] GPU platform. The systems were parameterized using the General Amber force field (GAFF) and AMBER 14 all-atom force field[62,63]. The non-bonded interactions were treated with a cut-off at 10 Å. Additionally, the Particle Mesh Ewald method[64] was used to compute long-range Coulomb interactions with a 10 Å nonbonded cut-off

for the direct space interactions. Energy minimization of the system was performed until a 10 kJ/mole tolerance energy was reached. Simulations were run using a 2 fs integration time step. The Langevin integrator[65] was used to maintain the system temperature at 300 K with a friction coefficient of 1 ps$^{-1}$. The initial velocities were assigned randomly to each atom using a Maxwell–Boltzmann distribution at 300 K. A cubic water box with a 10 Å padding to the nearest solute atom was filled by water molecules using the tip4p-Ew model[66]. The ionic strength of 0.1 M NaCl was applied, by adding the corresponding number of Na$^+$ and Cl$^-$ ions (specified later). Protonation states, equilibration protocols and other specifications for the individual system setups are described below. Production runs were performed under periodic boundary conditions, and trajectories were written every 10,000 steps (20 ps).

For the MD simulationen of NSD2 complexed with each peptide, the structure of human NSD2 (positions Y991–K1220) was modeled based on the cryo-EM structure of NSD2 E1099K, T1150A in complex with its nucleosome substrate (PDB: 7CRO)[51]. Reverting mutations of K1099E and A1150T were modeled using PyMOL (2.4.1)[67]. The missing part of the post-SET loop (positions P1206–K1220) was modeled based on the SET domain of SETD2 (PDB: 5V21)[68] using PyMOD 3.0[69], since no structure of NSD2 complexed with the H3K36 peptide and post-SET loop has been resolved. The histone tail of PDB 7CRO was replaced by the 15 aa long H3K36M peptide from PDB 5V21, and methionine 36 mutated to lysine. The H3K36 peptide (29-APATGGVKKPHRYRP-43) was manually mutated in PyMOL to generate the ssK36 peptide (29-APKTGGVKRPNNYRP-43). The K36 was manually deprotonated as required for the $S_N2$ mechanism[43,70,71]. AdoMet was modeled based on the coordinates in PDB 7CRO and parametrized using ANTECHAMBER from AmberTools (18.0)[72]. The Zn$^{2+}$ ions were modeled using the cationic dummy atom method[73–75]. Cysteines 1016, 1018, 1026, 1032, 1041, 1046, 1052, 1145, 1192, 1194, and 1199 were treated as unprotonated to ensure proper Zn$^{2+}$ binding[76]. The protein charge was neutralized and an ionic strength of 0.1 M NaCl was applied, by adding 30 Na$^+$ and 27 Cl$^-$ ions. Furhter information about the simulated systems are provided in Supplementary Table 6. To equilibrate the solvent, a 5 ns pressure coupled equilibration with Monte Carlo barostat[77] was performed at a pressure of 1 atm. Initially, the C-alpha (Cα) atoms of NSD2, the peptide, and the AdoMet atoms were restrained with a force constant of 100 and 5 kJ mol$^{-1}$ Å$^{-2}$, respectively. The restraints were removed successively, starting with the NSD2 Cα restraints, followed by a 5 ns equilibration with the peptide and AdoMet still being restrained. Subsequently, the AdoMet and peptide restraints were removed as well, followed by 5 ns equilibration with no restraints. For production, MD simulations were conducted in 50 replicates of 100 ns each (total simulation time 5 µs).

In order to define criteria describing a catalytically competent conformation, the following geometric requirements for a transition state (TS)-like conformation were derived from the known $S_N2$ geometry of methyl group transfer reaction[43,45] (Supplementary Fig. 7a).

(1) The distance between the lysine Nε and AdoMet methyl group C-atom is <4 Å.
(2) The angle between the lysine Nε, the lysine Cδ bond and the virtual bond between lysine Nε and the AdoMet methyl group C-atom is in a range of 109° ± 30°.
(3) The angle between the lysine Nε, the AdoMet methyl group C-atom and AdoMet S-atom bonds is in a range of 180° ± 30°.

Data analysis was performed utilizing MDTraj (1.9.4)[78] to calculate the distances and angles necessary for the geometric criteria of an $S_N2$ TS-like conformation. The RMSF analysis was conducted using MDTraj, where each trajectory was mapped to its first frame. All structures were visualized using PyMOL (2.4.1). The contact map analysis was performed utilizing contact map explorer (0.7.1)[46]. For the contact maps, a cut-off of 4.5 Å was used for the analysis. A contact was counted if at least one heteroatom of one residue was in a 4.5 Å$^3$ sphere surrounding one heteroatom from another residue excluding neighboring residues.

All PDB files and MD simulation protocols used in this study and additional MD simulation results obtained with peptides containing P30G and P43S changes as used for some of the peptide arrays are deposited at DaRUS (https://doi.org/10.18419/darus-3815).

## Statistics and reproducibility

The number of independent experimental repeats is indicated for each experiment. Standard deviations were determined with MS Excel. P-values were determined using two-sided T-test assuming unequal variance with MS Excel. Sample sizes are indicated in the text and figure legends.

## Reporting summary

Further information on research design is available in the Nature Portfolio Reporting Summary linked to this article.

## Data availability

All biochemical data generated or analyzed during this study are included in the published article and its supplementary files. Uncropped images of the Figures and Supplementary Figures are provided in Supplementary Fig. 12. The source data behind the graphs in the paper can be found in Supplementary Data 2. Additional information regarding the MD simulations, including the MD simulation checklist, modeled structures of NSD2 bound to different peptides and cofactors, starting structures of the MD runs, a movie of the MD run, and source data of the results of the MD analysis are provided on DaRUS (https://doi.org/10.18419/darus-3815). Plasmids generated in this work have been deposited with Addgene (84347). Any remaining information can be obtained from the corresponding author upon reasonable request.

## Code availability

MD simulations codes and analysis scripts are provided on DaRUS (https://doi.org/10.18419/darus-3815).

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

## Acknowledgements
This work was supported by the Deutsche Forschungsgemeinschaft under Germany's Excellence Strategy EXC 2075 390740016 in PN2-5.

## Author contributions
S.W. and A.J. devised the study. D.K. and S.W. conducted the biochemical experiments. P.S. conducted the MD simulations with the help of J.R. A.J. supervised the work. J.P. contributed to the supervision of the MD simulations. S.W., D.K., P.S., and A.J. prepared the figures. S.W., P.S., and A.J. wrote the manuscript draft. All authors were involved in data analysis and interpretation. The final manuscript was approved by all authors.

## Funding

## Competing interests
The authors declare no competing interests.
