## [Peer Review File · Communications Biology]

Reviewers' comments:

Reviewer #1 (Remarks to the Author):

In this manuscript, Weirich et al investigated the substrate sequence specificity of the lysine methyltransferase NSD2. The authors identified a new “super substrate” and use this information to identify two non-histone substrates that they confirm in cells. Molecular dynamics simulations are used to shed light on the mechanistic basis of NSD2 methylation of the super substrate. Overall, the data provided are high quality, and the study will be useful to the field. However, there are several issues that should be addressed by the reviewers:

1. In Figure 1, you show a representative image and the summary data of three independent experiments, mentioning “quantitatively analyzed, normalized and the data averaged.” Please provide details on the methods of how this one performed. It may be helpful to include the raw images of the two replicates in the supplement.
2. The authors perform a comparison of H3K36, H4K44, and H1.5 as NDS2 substrates but chose H3K36 as the basis of the subsequent experiments even though it had the lowest signal, why? Stronger methylation was also observed on histone H4 (Fig S1C).
 - a. No discussion of the sequences of H4 and H1.5 are included after performing experiments shown in Figure 1 and 2. Do any of the sequences in H4 and H1.5 correspond to observations made in subsequent experiments?
3. The presentation of data in Figure 2 is very hard to digest in the opinion of this reviewer. It might be helpful to label the figures with some of the sequences if possible. Another possible suggestion is moving Figure 2A and 2B into the supplement with the corresponding tables and labeling the sequences for Figure 2C in the main panel.
4. Is Figure 2D showing the corresponding Coomassie-stained gel with the gel used for autoradiography? It appears this is not the case since there is no NSD2-only lane in the Coomassie image. It is important to see the amount of NSD2 in each sample. Please include the Coomassie image from the autoradiography experiment. If this is not available, I suggest removing the Coomassie image into the supplement to avoid confusion.
5. Can the authors comment on the automethylation levels in figure 2D? Typically KMTs automethylate more when no substrate is present. This pattern is also usually observed when comparing substrates as well, for example in supplementary figure 1 the authors show that there is higher auto methylation levels for NSD2 in the presence of rec H3.1 compared to Rec H4, while the methylation on the substrates is reversed (higher on H4 than H3.1). However, in figure 2D the opposite is observed. Automethylation is consistent without any substrate and with the super substrate but lower with the wt H3K36 sequence.

The authors state in lines 208-210 that the next section will provide “mechanistic clues” but none are mentioned to explain the automethylation signal. In the discussion, the authors state in lines 358-260, “This observation suggests that H3K36 binds to the active site of NSD2, but it stays in a catalytically incompetent conformation, which blocks the active site and reduces automethylation.” How does this statement fit with the data shown in Figure S1C?

6. The authors estimate from the data shown in Figure 2d and corresponding images in supplemental Figure 5 that NSD2 methylates the super substrate 100-fold higher compared to WT H3K36. This statement seems inappropriate since there is no detectable methylation on the WT H3K36-GST protein in this case. Have the authors compared histone H3 as performed in Supplementary Figure 1? This 100-fold statement is repeated in the manuscript. It is clear that the super substrate is a more efficient substrate, but in this reviewer's opinion, the authors did not do enough experiments to substantiate claims of 100-fold improved.

7. Figure 3 provides summary data for what essentially becomes a binary – TS-like conformation or not. This is an important distinction, but also ignores representing the data in a more clear way. Could the authors provide plots summarizing the three TS-like conformation parameters used showing the time in each category?

8. Analysis of the MD simulations lead the authors to hypothesize that NSD2 in complex with the super substrate stabilizes the interactions with AdoMet (lines 271-273). Can the authors test this hypothesis?

9. Please show the data for other tested protein substrates. The authors show purification of 17 putative protein substrates and state 2 showed signal (shown in figure 6C), but the data for the remaining 15 is missing.

a. Furthermore, can the authors comment on why the 15 were not methylated? Does this correlate with signal on peptides? Does structural information (if available or alpha fold predictions) shed light on the two that were successful?

10. Have the authors compared the newly identified substrates to H3 protein or nucleosome substrates? This comparison should be included and shown, preferably on the same autoradiograph. The authors cited guidelines (Ref 11) include this recommendation as Rule #1.

Reviewer #2 (Remarks to the Author):

The manuscript by Sara Weirich et al. studies the specificity of protein lysine methyltransferase NSD2 with respect to different substrates, including its native substrate -

H3K36 site in histone H3, certain artificial substrates and other nuclear proteins. Using experimental SPOT analysis the authors revealed amino acids in the vicinity of H3K36 that are important for targeting by NSD2, designed a super-substrate that is methylated at a considerably higher rate and identified targets of NSD2 in other nuclear proteins. Using MD simulations the authors studied the process of NSD2 interaction with its substrates, formation of transition states and demonstrated that the increase in activity for the super-substrate is due to distinct hyperactive conformations of the enzyme-peptide complex. Overall, I have a positive impression about this study, it includes an elegant combination of experimental and theoretical treatment of the addressed problem. H3K36 is an important histone PTM site, its mutations are implicated in several cancers. This study enhances our understanding of the dynamic mechanism involved in epigenetic regulation of gene expression and DNA repair. However, I feel that certain improvements have to be made to the text and the presentation of the results before this study will be ready to be published.

Major:

1) The presented work is among a series of studies by the same and other authors on the specificity of different PKMTs to the H3K36 site, including the design of super-substrates for SETD2 (eg., refs. 43, 44). No structural analysis or discussion is provided to comprehend why the super-substrates for SETD2 and NSD2 are different, and how this may be related to the evolution or functional specificity of different PKMTs. Such an addition will clearly make the paper more interesting to the general readership.

2) I'm not very much happy with the way how MD simulations results and methods are presented.

2.1) From the text I got the impression that the only analysis of conformational dynamics and its changes between the native and super-substrate peptide revealed by MD simulations is presented in Figure 3a. While Figure 5 (that is meant to show in detail what conformational changes and interaction changes happen) is based on the starting structures (not very clear what are those). Hence, much of the results in section "Contact analysis of H3K36 and ssK36-NSD2 complexes" look like conjectures about how certain interactions might change the geometry of the complex, rather than showing directly from the simulations how certain aspects of geometry have changed. I'd suggest to also clearly show in the Figures how Sn2 TS-like conformations look like, and what are the differences between those conformation and conformations that does not look like Sn2.

2.2) The methods of MD simulations should be described in detail to allow for reproducibility. Leaving such important information as the force fields used for simulations, simulation software as references to other papers generally does not

contribute to reproducibility. Please, address explicitly points 3b, 4a, 4b, 4c of the “Reliability and reproducibility checklist for molecular dynamics simulations”. What force field was used, what water model was used, why those parameters were used, what simulation engine was employed, what were the cut-off parameters, etc. The table with the information about the simulated systems should be provided. I would strongly advocate for providing the Movies and trajectory files as supplementary information.

3) Related to point 2, supplementary information should be clearly described and referenced in the text. Currently, the link in the manuscript <https://doi.org/10.18419/darus-3263> points to the SI of another study (by Khella et al.) While looking at one of the videos in that repository I was surprised that AdoMet (?) molecule was moving around during the simulations, is it the same case in the current study? Can it affect the conformation of the peptide?

Minor:

1. English editing is needed in the manuscript. Please, use consistently US or UK spelling. US spelling: artifact, modeling, tumor, etc. Numbers below ten are usually spelled out as words. “50 replicates à 100 ns “ => “50 replicates of 100 ns each”. L. 296 “in the presence”. L. 355 “bent”. L. 355 “help K36 approach ”. l. 185 “designed” => “designated”. L. 30 “is increased”. “amino acid exchanges” => “amino acid changes or substitutions”, “MD simulation”=>”MD simulations”
2. The caption of Figure 5 contains “Overlay and starting structure...” is unclear.
3. In different experiments the authors used different exposure times for the autoradiographic image, please add some details about the choice of these times.
4. It is unclear why there are no signals for the wildtype GST-tagged H3K36 protein (lines 198-210).
5. Line 285 (www.proteinatlas.org/ENSG00000109685-NSD2): please, add version of Protein Atlas, because the link may become inactive after a while

Reviewer #3 (Remarks to the Author):

In this manuscript, Jeltsch and coworkers described the investigation of the substrate sequence specificity of the human protein lysine methyltransferase NSD2. Using peptide SPOT array methylation assay, the authors observed that amino acid residues different from the natural ones in the H3K36 target were preferred at some positions in the specificity profile. Thus, they combined four of these preferred residues to yield a super-substrate which was methylated at least 100-fold faster by NSD2 at peptide and protein

level. Using molecular dynamics simulations, they demonstrated that this activity increase is due to distinct hyperactive conformations of the enzyme-peptide complex. Then, they conducted a proteome wide search for nuclear proteins matching the specificity profile of NSD2 which led to the discovery of 22 novel peptide substrates. After cloning the corresponding non-histone substrate candidates with GST-tag and expressing them in *E. coli*, protein methylation studies led to the identification of K1033 of ATP-dependent helicase (ATRX) and K819 of Fanconi anemia group M (FANCM) protein as novel NSD2 protein substrates. The methylation was confirmed in human cells.

In general, the manuscript is very well written and organized, the experiments cleverly designed and cunningly executed, and the conclusions are sound and consistent with the results. In my opinion, the results of these studies are particularly noteworthy as they strengthen the connection of NSD2 and H3K36 methylation to DNA repair.

Therefore, I recommend the manuscript for publication in *Communications Biology*, provided that the authors will address a few minor issues:

- 1) Besides MD simulation studies, the authors should assess the affinity constant (KD) as well as kinetic association and dissociation constants (and residence time) of the binding of the super-substrate peptide ssK36 to NSD2 using biophysical techniques (DSF, SPR, ITC, and so on) and compare them to those obtained for the canonical H3K36 peptide. This would certainly help the discussion of the results they obtained.
- 2) I was really bothered by having to go back and forth in the manuscript to give a look at figures and tables. Please, embed figures, charts, tables, schemes, and equations in the text at the point of relevance.

Reviewer #4 (Remarks to the Author):

NSD2 is an important H3K36 methyltransferase. In the manuscript, authors profiled the substrate sequence specificity of NSD2 and discovered a preferred super-substrate sequence, which was methylated >100-fold faster by NSD2 than WT H3 peptide. Authors further used molecular dynamics simulations to demonstrate how the super-substrate sequence peptide induces the hyperactive conformations of the enzyme-peptide complex. A search for human nuclear proteins matching the NSD2 specificity profile led to the discovery of 22 novel peptide substrates, in which ATRX and FANCM were further verified to be NSD2 substrates in vitro and in human cells. The data are presented clearly and convincingly. I would, in principle, recommend publication. However, I have some minor comments.

1. Because the NSD2 target site might be buried in folded proteins, to prove the non-histone substrates are methylated by NSD2 at protein level, authors purified these proteins and tested the methylation of these proteins by NSD2. How many of these tested proteins are full-length proteins? And how many of these are just a truncated fragment? I noticed that both ATRX and FANCM, the two substrates authors further verified, are not full-length proteins. Only a small portion of ATRX or FANCM was tested in vitro and in mammalian cells. I can understand that both ATRX and FANCM are very large proteins and it is probably infeasible to purify the full-length proteins. Since authors did not really test the methylation of ATRX or FANCM in full-length protein level, authors should at least discuss the limitation of this study. If additional experiments could be carried out to analyze the methylation of ATRX and FANCM by NSD2 at full-length and cellular level, it will be big plus for this manuscript.

2. The same strategy has been used by the same lab to identify the super-substrate for SETD2. Interestingly, as shown in Figure S4, NSD2 did not catalyze the methylation of SETD2 super-substrate at all. The sequences of NSD2 super-substrate and SETD2 super-substrate are very similar and only T32/F32 and N40/R40 positions are different, but they show dramatic different activity. Thus, additional structural analyses or mutagenesis studies are desired to show why these two super-substrates are differently recognized by NSD2 and SETD2. Guided by these information, authors could identify NSD2-biased and SETD2-biased non-histone substrates, which will be of broad interest to the readers of Communication Biology.

3. To identify novel substrates of NSD2, authors used ScanSite server to search with a sequence motif which does not strictly follow super-substrate sequence. As a result, the identified sequences from database (e.g. ATRX and FANCM) are far different from the super-substrate sequence (as shown below). I wonder whether there is a sequence nearly-perfectly matching the super-substrate sequence (probably just one or two position varied) and whether this ssK36-like sequence could be methylated more efficiently by NSD2 than ATRX or FANCM.

ATRX: CHFPKGKQIKNGTT

FANCM:HKKSSFIKNINQGSS

ssK36:APKTGGVKKRPNNYRP

H3K36:APATGGVKKPHRYRP

4. Authors claimed that H3K36 is methylated at least 100-fold weaker than super-substrate ssK36. This estimation might be too rough. Is that possible to measure the kinetic parameters for NSD2-catalyzed reaction to get a more accurate number?

Reply to the reviewers' comments

Reply to all reviewers: Thank you very much for your careful work with our manuscript and the insightful comments and suggestions which have helped us to further improve the quality of our manuscript. We have addressed all your comments in the revision.

Reviewer #1

"In this manuscript, Weirich et al investigated the substrate sequence specificity of the lysine methyltransferase NSD2. The authors identified a new "super substrate" and use this information to identify two non-histone substrates that they confirm in cells. Molecular dynamics simulations are used to shed light on the mechanistic basis of NSD2 methylation of the super substrate. Overall, the data provided are high quality, and the study will be useful to the field. However, there are several issues that should be addressed:"

Reply: Thank you very much for the overall positive assessment. We have carefully addressed and/or answered all your questions and comments in our revision of the manuscript.

"1. In Figure 1, you show a representative image and the summary data of three independent experiments, mentioning "quantitatively analyzed, normalized and the data averaged." Please provide details on the methods of how this one performed. It may be helpful to include the raw images of the two replicates in the supplement."

Reply: We have now explained the data analysis in the Method section in more details. Normalization refers to the rescaling of the data obtained in the individual experiments between 0 and 1, using the highest and lowest activity of the corresponding membrane. This clarifies the procedure.

*"2. The authors perform a comparison of H3K36, H4K44, and H1.5 as NDS2 substrates but chose H3K36 as the basis of the subsequent experiments even though it had the lowest signal, why? Stronger methylation was also observed on histone H4 (Fig S1C).
a. No discussion of the sequences of H4 and H1.5 are included after performing experiments shown in Figure 1 and 2. Do any of the sequences in H4 and H1.5 correspond to observations made in subsequent experiments?"*

Reply: It has been shown that H4K44 is not methylated in cells (Li et al., 2009). For H1.5K168, no information about cellular methylation is available, hence we decided to continue our study with H3K36 as the only validated cellular histone target of NSD2. The difference in methylation levels of H3K36 and H4K44 is not very strong when considering the different experiments shown in Supplementary Figure 1B and the concentrations of the H3 and H4 proteins in Supplementary Figure 1C. The elevated peptide methylation level of H1.5K168 peptides (when compared with H3K36) cannot be explained on the basis of the single mutation analysis conducted here. Hence, they must result from the combined effects of two or more exchanges. This information is now provided in the newly added "Limitations of the study" section at the end of the discussion. We have now added a new experiment, showing the methylation H3K36, H4K44, H1.5K168 und ssK36 on one array which illustrates that ssK36 is much better methylated even than H1.5K168 (Supplementary Figure 3B).

“3. The presentation of data in Figure 2 is very hard to digest in the opinion of this reviewer. It might be helpful to label the figures with some of the sequences if possible. Another possible suggestion is moving Figure 2A and 2B into the supplement with the corresponding tables and labeling the sequences for Figure 2C in the main panel.”

Reply: Thank you. Figure 2 has been reorganized along the lines proposed by you.

“4. Is Figure 2D showing the corresponding Coomassie-stained gel with the gel used for autoradiography? It appears this is not the case since there is no NSD2-only lane in the Coomassie image. It is important to see the amount of NSD2 in each sample. Please include the Coomassie image from the autoradiography experiment. If this is not available, I suggest removing the Coomassie image into the supplement to avoid confusion.”

Reply: It is technically not possible for us to show the Coomassie stain and radioactivity from the same gel in framework of rules for radioactive work at our institution. The gels were loaded and run in parallel. This has now been clarified in the figure legend. We still believe showing the Coomassie gel is helpful for the reader. This is now shown in Supplementary Figure 5A.

“5. Can the authors comment on the automethylation levels in figure 2D? Typically KMTs automethylate more when no substrate is present. This pattern is also usually observed when comparing substrates as well, for example in supplementary figure 1 the authors show that there is higher auto methylation levels for NSD2 in the presence of rec H3.1 compared to Rec H4, while the methylation on the substrates is reversed (higher on H4 than H3.1). However, in figure 2D the opposite is observed. Automethylation is consistent without any substrate and with the super substrate but lower with the wt H3K36 sequence. The authors state in lines 208-210 that the next section will provide “mechanistic clues” but none are mentioned to explain the automethylation signal. In the discussion, the authors state in lines 358-260, “This observation suggests that H3K36 binds to the active site of NSD2, but it stays in a catalytically incompetent conformation, which blocks the active site and reduces automethylation.” How does this statement fit with the data shown in Figure S1C?”

Reply: Thank you for pointing out this issue. However, in Fig. 2E smaller amounts of the super-substrate were used. In the experiments with H3 and H4, same weight concentrations were used, which however translate into different molar concentrations. Hence, the effects of the substrates on the NSD2 automethylation cannot be clearly unraveled and we have removed the corresponding statements in our manuscript. We apologize for the confusion caused. This is now shown in Supplementary Figure 5A.

“6. The authors estimate from the data shown in Figure 2d and corresponding images in supplemental Figure 5 that NSD2 methylates the super substrate 100-fold higher compared to WT H3K36. This statement seems inappropriate since there is no detectable methylation on the WT H3K36-GST protein in this case. Have the authors compared histone H3 as performed in Supplementary Figure 1? This 100-fold statement is repeated in the manuscript. It is clear that the super substrate is a more efficient substrate, but in this reviewer's opinion, the authors did not do enough experiments to substantiate claims of 100-fold improved.”

Reply: Thank you for highlighting this point. We have repeated methylation analyses using ssK36 and H3K36 peptides as well as the GST-tagged H3K36 and ssK36 protein substrates in new experiments now shown in Figure 2D and 2E. Based on dilution series of the methylated ssK36 peptide or protein loaded on the same gel together with the corresponding H3K36 sample, we can now determine that ssK36 is methylated 91-fold more readily at the peptide level and about 3000-fold better at protein level than H3K36. This point has been explained in the text now on p. 6-7. The description of the fold-stimulation has been adjusted correspondingly throughout.

“7. Figure 3 provides summary data for what essentially becomes a binary – TS-like conformation or not. This is an important distinction, but also ignores representing the data in a more clear way. Could the authors provide plots summarizing the three TS-like confirmation parameters used showing the time in each category?”

Reply: The condition for a potential TS-like structure is that all geometric constraints are fulfilled at the same time. Based on this, we respectfully like to mention that analysis and documentation of each individual parameter would not be very helpful. It would rather be confusing, because one parameter could be satisfied in conformations that fail to comply entirely with another parameter. These conformations could not contribute to “in silico” activity and, hence, including them in averages or compiled presentations would be strongly misleading.

“8. Analysis of the MD simulations lead the authors to hypothesize that NSD2 in complex with the super substrate stabilizes the interactions with AdoMet (lines 271-273). Can the authors test this hypothesis?”

Reply: Thank you for this comment. We now show in Supplementary Figure 7B that the fluctuations of the AdoMet heteroatoms are strongly reduced in the NSD2-ssK36-AdoMet complex, as compared to NSD2-H3K36-AdoMet. This observation suggests that the AdoMet is stabilized in its catalytically competent conformation in NSD2-ssK36-AdoMet by the additional contacts which are specific for this complex.

“9. Please show the data for other tested protein substrates. The authors show purification of 17 putative protein substrates and state 2 showed signal (shown in figure 6C), but the data for the remaining 15 is missing.”

Reply: These are just autoradiographic images of gels not showing substrate methylation signals. All of them show NSD2 automethylation as a kind of “positive control” ensuring the technical correctness. All experiments were conducted at least twice. We do not think that it will be helpful for readers to show all negative images.

“a. Furthermore, can the authors comment on why the 15 were not methylated? Does this correlate with signal on peptides? Does structural information (if available or alpha fold predictions) shed light on the two that were successful?”

Reply: Thank you very much for this insightful proposal. We have made the requested analysis which is now shown in the new Supplementary Table 6. It revealed that indeed ATRX and FANCM were the only proteins containing the target K in a region shown or predicted to be unfolded. This suggests

that NSD2 cannot access target peptides that form a secondary structure. This information is now mentioned in the results and discussion section and we also discuss that in cells NSD2 might approach its targets during the protein folding process or with the help of chaperones in the “Limitations of the study” section.

“10. Have the authors compared the newly identified substrates to H3 protein or nucleosome substrates? This comparison should be included and shown, preferably on the same autoradiograph. The authors cited guidelines (Ref 11) include this recommendation as Rule #1.”

Reply: Thank you for this comment. We now provide additional gels in Supplementary Figure 9 which allow to relate the methylation of ATRX and FANCM to H3. We also mention on p. 12 now, that the methylation of ATRX is comparable to H3 while methylation of FANCM is weaker.

Reviewer #2 (Remarks to the Author)

“The manuscript by Sara Weirich et al. studies the specificity of protein lysine methyltransferase NSD2 with respect to different substrates, including its native substrate - H3K36 site in histone H3, certain artificial substrates and other nuclear proteins. Using experimental SPOT analysis the authors revealed amino acids in the vicinity of H3K36 that are important for targeting by NSD2, designed a super-substrate that is methylated at a considerably higher rate and identified targets of NSD2 in other nuclear proteins. Using MD simulations the authors studied the process of NSD2 interaction with its substrates, formation of transition states and demonstrated that the increase in activity for the super-substrate is due to distinct hyperactive conformations of the enzyme-peptide complex. Overall, I have a positive impression about this study, it includes an elegant combination of experimental and theoretical treatment of the addressed problem. H3K36 is an important histone PTM site, its mutations are implicated in several cancers. This study enhances our understanding of the dynamic mechanism involved in epigenetic regulation of gene expression and DNA repair. However, I feel that certain improvements have to be made to the text and the presentation of the results before this study will be ready to be published.”

Reply: Thank you very much for the very positive overall assessment. We have carefully addressed and/or answered all your questions and comments in our revision of the manuscript.

“Major:

1) The presented work is among a series of studies by the same and other authors on the specificity of different PKMTs to the H3K36 site, including the design of super-substrates for SETD2 (eg., refs. 43, 44). No structural analysis or discussion is provided to comprehend why the super-substrates for SETD2 and NSD2 are different, and how this may be related to the evolution or functional specificity of different PKMTs. Such an addition will clearly make the paper more interesting to the general readership.”

Reply: Thank you for raising this very interesting question. However, the similarity of the two peptide interfaces of NSD2 and SETD2 is much lower than one might anticipate. Below a part of the multiple sequence alignment of NSD enzymes and SETD2 is shown with the residues highlighted that were shown to be involved in a differential interaction with the H3K36 and ssK36 substrates in NSD2 (blue shade, data from the current manuscript) or SETD2 (red shade, data are taken from Schuhmacher et

al., 2020). Strikingly, only 12 out of the 42 NSD2 residues (29%) are identical between NSD2 and SETD2 indicating that the peptide interaction is very different in its molecular details in both cases.

```

2300          *          2320          *          2340          *          2360
SETD2 huma : QCECTPLSKDERAQQEITACG--EDCLNRLMLIECSS-RCFNGDYCSNRRFQRKQHADVVEVILTEKKGWGLRAAK : 1568
SETD2 mous : QCECTPLSKDERAQQEVACG--EDCLNRLMLIECSS-RCFNGDYCSNRRFQRKQHADVVEVILTEKKGWGLRAAK : 1542
NSD2 human : KCNCKPT-----DENPCGFDSECLNRMLMFECHPQVCPAGEYFCQNQCFTKROYPETKIIKTDGKGWGLVAKR : 1081
NSD2 mouse : KCNCKPT-----DENPCGSDSECLNRMLMFECHPQVCPAGEYFCQNQCFTKROYPETKIIKTDGKGWGLVAKR : 1081
NSD1 human : RCNCKAT-----DENPCGIDSECLNRMLLYECHFTVCPAGGRCONQCFSKROYPEVEIFRITLQRGWGLRTKT : 1960
NSD1 mouse : RCNCKAT-----DENPCGIDSECLNRMLLYECHFTVCPAGVRCONQCFSKROYPDVEIFRITLQRGWGLRTKT : 1858
NSD3 human : RCNCKEA-----DENPCGLESECLNRMLQYECHEPQVCPAGDRCONQCFTRKLYPDAEIIKTERRGWGLRTKR : 1163
NSD3 mouse : RCNCKEG-----DENPCGLESOCLNRMSQYECHEPQVCPAGDRCONQCFTRKLYPDAEVIKTERRGWGLRTKR : 1166

```

```

*          2380          *          2400          *          2420          *          2440
SETD2 huma : DLPSNTFVLEYCGEVLVDHKEFKARVREYARLNKNIHYFMAKNDIIDATQKGNCSRFMNHSCPN CETOKWTV : 1642
SETD2 mous : DLPSNTFVLEYCGEVLVDHKEFKARVREYARLNKNIHYFMAKNDIIDATQKGNCSRFMNHSCPN CETOKWTV : 1616
NSD2 human : DIRKGEFVNEYV GELIDEEEC MARIKYAHENDITHEFYMLTIDKDRIIDAGPKGNYSRFMNHSCPN CETOKWTV : 1155
NSD2 mouse : DIRKGEFVNEYV GELIDEEEC MARIKYAHENDITHEFYMLTIDKDRIIDAGPKGNYSRFMNHSCPN CETOKWTV : 1155
NSD1 human : DIKKGEFVNEYV GELIDEEECRARIRRYAQEHDTNFYMLTIDKDRIIDAGPKGNYSRFMNHSCPN CETOKWTV : 2034
NSD1 mouse : DIKKGEFVNEYV GELIDEEECRARIRRYAQEHDTNFYMLTIDKDRIIDAGPKGNYSRFMNHSCPN CETOKWTV : 1932
NSD3 human : SIKKGEFVNEYV GELIDEEECRLRIKRAHENSVINFYMLTIVTKDRIIDAGPKGNYSRFMNHSCPN CETOKWTV : 1237
NSD3 mouse : SIKKGEFVNEYV GELIDEEECRLRIKRAHENSVINFYMLTIVTKDRIIDAGPKGNYSRFMNHSCPN CETOKWTV : 1240

```

```

*          2460          *          2480          *          2500          *
SETD2 huma : NGQLRVGFETTKLVPSGSELTFDYQFQRYGKEAQKCFCCSANCRCGYLGGENRVSIRAAAGGKMKKERSR--KKDS : 1714
SETD2 mous : NGQLRVGFETTKLVPSGSELTFDYQFQRYGKEAQKCFCCSANCRCGYLGGENRVSIRAAAGGKMKKERSR--KKDS : 1688
NSD2 human : NGDTRVGLFVAFCDIPAGTELTFNYNLDCLNGEKTVCRCGASNCSSGFLGDRPKTSTTSSEEGKTKKTRRRR : 1229
NSD2 mouse : NGDTRVGLFVAFCDIPAGTELTFNYNLDCLNGEKTVCRCGASNCSSGFLGDRPKTSTTSSEEGKTKKTRRRR : 1229
NSD1 human : NGDTRVGLFALSIDIKAGTELTFNYNLECLNGKTVCKCGAPNCSGFLGVRPKNQPIATEEKSRRKFKKQQQKRR : 2108
NSD1 mouse : NGDTRVGLFALSIDIKAGTELTFNYNLECLNGKTVCKCGAPNCSGFLGVRPKNQPIVTEEKSRRKFKRPHGKRR : 2006
NSD3 human : NGDTRVGLFALCDIPAGMELTFNYNLDCLNGRTVCHCGADNCSGFLGVRPKSACASTNEEKAKNAKLRKQKRRK : 1311
NSD3 mouse : NGDTRVGLFALCDIPAGMELTFNYNLDCLNGRTVCHCGADNCSGFLGVRPKSACTSAVDEKTKNAKLRKQKRRK : 1313

```

This divergence more specifically manifests in the peptide interaction of both enzymes at the 3 residues differing between both super-substrates: at the -4 site (R vs. K), at the -3 site (T vs. F), and at the +4 site (N vs. R). The SETD2 and NSD2 residues preferentially interacting with these substrate residues are listed in the following table.

Pos.	SETD2		NSD2	
	ssK36 sequence	interacting residues	ssK365 sequence	interacting residues
-4	R	E1674	K	I1114 (not conserved) T1115 (not conserved) N1186 (not conserved) E1187 K1188 (not conserved) T1189 (not conserved)
-3	F	E1674 A1675 (not conserved) Q1676 (not conserved)	T	T1115(not conserved) H1116 F1117 (not conserved) L1181 (not conserved) C1183 (not conserved) K1188 (not conserved) T1189 (not conserved)
+4	R	M1526 Q1638 (not conserved)	N	N1178 (not conserved)

Evidently, from the 20 amino acids involved in the recognition of these 3 substrate residues in both enzymes, only one is involved in a comparable interaction in both cases (E1674/E1187 interacting with the -4 residue), but even this residue is engaged in different contact patterns to the -3 substrate residue. Hence, depicting the molecular differences between the substrate interaction of NSD2 and SETD2 super-substrates will be a highly complicated task, that certainly is beyond the scope of what can be done in a revision of this work.

“2) I’m not very much happy with the way how MD simulations results and methods are presented. 2.1) From the text I got the impression that the only analysis of conformational dynamics and its changes between the native and super-substrate peptide revealed by MD simulations is presented in Figure 3a. While Figure 5 (that is meant to show in detail what conformational changes and interaction changes happen) is based on the starting structures (not very clear what are those). Hence, much of the results in section “Contact analysis of H3K36 and ssK36-NSD2 complexes” look like conjectures about how certain interactions might change the geometry of the complex, rather than showing directly from the simulations how certain aspects of geometry have changed. I’d suggest to also clearly show in the Figures how Sn2 TS-like conformations look like, and what are the differences between those conformation and conformations that does not look like Sn2.”

Reply: We apologize for not being clear. Figure 3A just shows some example peptide structures in the context of one NSD2 structure just for illustration. Fig. 3B and C show the analysis of the in silico activity taken from all MD simulations. Figure 4 compares the intensity of contact along the entire simulation and identifies differences. The following description and Fig. 5 aim to visualize the effects observed in the MD simulations. This is done using example structural snapshots that illustrate the corresponding effect. This all has now been explained better in the corresponding figure legends.

“2.2) The methods of MD simulations should be described in detail to allow for reproducibility. Leaving such important information as the force fields used for simulations, simulation software as references to other papers generally does not contribute to reproducibility. Please, address explicitly points 3b, 4a, 4b, 4c of the “Reliability and reproducibility checklist for molecular dynamics simulations”. What force field was used, what water model was used, why those parameters were used, what simulation engine was employed, what were the cut-off parameters, etc. The table with the information about the simulated systems should be provided. I would strongly advocate for providing the Movies and trajectory files as supplementary information.”

Reply: The requested information has been added to the method description and the MD simulation checklist. The system information table has been added as new Supplementary Table 7. A movie has been added to the Darus repository. The data deposited in Darus including modelled structures of NSD2 bound to different peptides, starting structures of the simulations, source data of the results of the MD analysis, MD simulations codes and analysis scripts will allow anyone to easily reproduce the simulation. We consider this as a more useful and resource efficient way of data sharing than uploading the very huge trajectory files.

“3) Related to point 2, supplementary information should be clearly described and referenced in the text. Currently, the link in the manuscript <https://doi.org/10.18419/darus-3263> points to the SI of another study (by Khella et al.) While looking at one of the videos in that repository I was surprised

that AdoMet (?) molecule was moving around during the simulations, is it the same case in the current study? Can it affect the conformation of the peptide?"

Reply: The given reviewer link given in the paper was correct and working. Unfortunately, there was a mistake in the Darus number that was mentioned. The wrong DOI citation has been corrected. We like to apologize for this stupid error. The Darus entry has been released now, such that no reviewer login is required any more.

Certainly, AdoMet can move in the MD simulations as all other parts of the NSD2-peptide-AdoMet complex can do. This is now also shown in the newly provided movie file. Perhaps there was a confusion with the wrong Darus link. In the previous study related to the wrong link, AdoMet sMD simulations were performed, in which the association process of AdoMet into the PKMT-peptide complex had been investigated. In the current study, we are not concerned with this but only investigate the dynamic movement of all components in the ternary NSD2-peptide-AdoMet complex.

"Minor:

1. English editing is needed in the manuscript. Please, use consistently US or UK spelling. US spelling: artifact, modeling, tumor, etc. Numbers below ten are usually spelled out as words. "50 replicates à 100 ns " => "50 replicates of 100 ns each". L. 296 "in the presence". L. 355 "bent". L. 355 "help K36 approach ". l. 185 "designed" => "designated". L. 30 "is increased". "amino acid exchanges" => "amino acid changes or substitutions", "MD simulation"=>"MD simulations""

Reply: Thank you for these detailed corrections. We have carefully checked for US spelling. Your corrections have been adopted except the proposal to spell out numbers as this would diminish readability (e.g, "- one site" or "minus one site", instead of "-1 site").

"2. The caption of Figure 5 contains "Overlay and starting structure..." is unclear."

Reply: This has been rewritten for clarity.

"3. In different experiments the authors used different exposure times for the autoradiographic image, please add some details about the choice of these times."

Reply: Exposure times were chosen based on previous experience, but also following the weekly and daily working schedules. As the exposure times are always indicated and we do not refer to comparisons across different images, this is sufficiently documented.

"4. It is unclear why there are no signals for the wildtype GST-tagged H3K36 protein (lines 198-210)."

Reply: Thank you for highlighting this point. The absence of methylation on the H3K36 sample was due to relatively low enzyme and substrate concentrations. We have repeated methylation analyses using ssK36 and H3K36 peptides as well as the GST-tagged H3K36 and ssK36 protein substrates in new experiments with adjusted conditions now shown in Figure 2D and 2E. In both reactions, activity on H3K36 was observed. Based on dilution series of the methylated ssK36 peptide or protein loaded on the same gel together with the corresponding H3K36 sample, we can now determine that ssK36 is methylated 91-fold more readily at the peptide level and about 3000-fold better at protein level than

H3K36. This point has been explained in the text now on p. 6-7. The description of the fold-stimulation has been adjusted correspondingly throughout.

"5. Line 285 (www.proteinatlas.org/ENSG00000109685-NSD2): please, add version of Protein Atlas, because the link may become inactive after a while"

Reply: Thank you very much. This has been added as requested.

Reviewer #3

"In this manuscript, Jeltsch and coworkers described the investigation of the substrate sequence specificity of the human protein lysine methyltransferase NSD2. Using peptide SPOT array methylation assay, the authors observed that amino acid residues different from the natural ones in the H3K36 target were preferred at some positions in the specificity profile. Thus, they combined four of these preferred residues to yield a super-substrate which was methylated at least 100-fold faster by NSD2 at peptide and protein level. Using molecular dynamics simulations, they demonstrated that this activity increase is due to distinct hyperactive conformations of the enzyme-peptide complex. Then, they conducted a proteome wide search for nuclear proteins matching the specificity profile of NSD2 which led to the discovery of 22 novel peptide substrates. After cloning the corresponding non-histone substrate candidates with GST-tag and expressing them in E. coli, protein methylation studies led to the identification of K1033 of ATP-dependent helicase (ATRX) and K819 of Fanconi anemia group M (FANCM) protein as novel NSD2 protein substrates. The methylation was confirmed in human cells.

In general, the manuscript is very well written and organized, the experiments cleverly designed and cunningly executed, and the conclusions are sound and consistent with the results. In my opinion, the results of these studies are particularly noteworthy as they strengthen the connection of NSD2 and H3K36 methylation to DNA repair.

Therefore, I recommend the manuscript for publication in Communications Biology, provided that the authors will address a few minor issues:"

Reply: Thank you very much for the very positive overall assessment. We have carefully addressed and/or answered both of your questions and comments in our revision of the manuscript.

"1) Besides MD simulation studies, the authors should assess the affinity constant (KD) as well as kinetic association and dissociation constants (and residence time) of the binding of the super-substrate peptide ssK36 to NSD2 using biophysical techniques (DSF, SPR, ITC, and so on) and compare them to those obtained for the canonical H3K36 peptide. This would certainly help the discussion of the results they obtained."

Reply: Thank you very much for this comment. We agree that information about the binding of the peptides and AdoMet would be interesting. However, we like to mention that our work addresses enzymatic catalysis. For this stabilization of the TS is the only relevant parameter, not ground state binding of substrates as determined in thermodynamic binding assays like DSR, SPR or ITC. In fact, stable ground state binding of substrates is not beneficial for enzyme catalysis from a theoretical point of view. We directly investigate the stabilization of the TS of the NSD2 methylation reaction by two approaches, single turnover kinetics and molecular dynamic simulations. In the kinetics the single turnover rate constant is measured which is directly related to the TS energy. In the MD

simulations, the accessibility of the TS is analyzed which also provides direct information about its energy. To obtain information about the binding of substrate peptides and AdoMet under kinetically relevant conditions, steady-state kinetics would be appropriate allowing to determine the corresponding K_m -values. However, given the low in vitro activity of NSD2, technically sound steady-state kinetics cannot be conducted, because they require an excess of substrate over enzyme (common protocols refer to at least 5-fold). This issue has now been mentioned in the new “Limitations of the Study” section of our manuscript.

“2) I was really bothered by having to go back and forth in the manuscript to give a look at figures and tables. Please, embed figures, charts, tables, schemes, and equations in the text at the point of relevance.”

Reply: We are sad to hear this. Different journals and scientists have different preferences where to place figures. One argument for putting them at the end is that this allows at least to find them easily, while one has to search if they are distributed throughout the paper. We have inspected the submission guidelines of Comm. Biol. and realized that they concur with your recommendations. Hence, we have now placed figures in the manuscript at their approximate places of insertion.

Reviewer #4

“NSD2 is an important H3K36 methyltransferase. In the manuscript, authors profiled the substrate sequence specificity of NSD2 and discovered a preferred super-substrate sequence, which was methylated >100-fold faster by NSD2 than WT H3 peptide. Authors further used molecular dynamics simulations to demonstrate how the super-substrate sequence peptide induces the hyperactive conformations of the enzyme-peptide complex. A search for human nuclear proteins matching the NSD2 specificity profile led to the discovery of 22 novel peptide substrates, in which ATRX and FANCM were further verified to be NSD2 substrates in vitro and in human cells. The data are presented clearly and convincingly. I would, in principle, recommend publication. However, I have some minor comments.”

Reply: Thank you very much for the very positive overall assessment. We have carefully addressed and/or answered all your questions and comments in our revision of the manuscript.

“1. Because the NSD2 target site might be buried in folded proteins, to prove the non-histone substrates are methylated by NSD2 at protein level, authors purified these proteins and tested the methylation of these proteins by NSD2. How many of these tested proteins are full-length proteins? And how many of these are just a truncated fragment? I noticed that both ATRX and FANCM, the two substrates authors further verified, are not full-length proteins. Only a small portion of ATRX or FANCM was tested in vitro and in mammalian cells. I can understand that both ATRX and FANCM are very large proteins and it is probably infeasible to purify the full-length proteins. Since authors did not really test the methylation of ATRX or FANCM in full-length protein level, authors should at least discuss the limitation of this study. If additional experiments could be carried out to analyze the methylation of ATRX and FANCM by NSD2 at full-length and cellular level, it will be big plus for this manuscript.”

Reply: We have now added a new supplemental table providing all details about the cloned proteins including borders of the cloned regions (Supplementary Table 6). We apologize for not providing this

information in the previous version of the paper. Indeed, most of the proteins could not be cloned and purified in full-length form. We have added a statement to the limitations of the study indicating that methylation experiments were conducted with protein fragments of ATRX and FANCM and methylation should be further studied with the full-length, endogenous proteins. Thank you for pointing this out.

“2. The same strategy has been used by the same lab to identify the super-substrate for SETD2. Interestingly, as shown in Figure S4, NSD2 did not catalyze the methylation of SETD2 super-substrate at all. The sequences of NSD2 super-substrate and SETD2 super-substrate are very similar and only T32/F32 and N40/R40 positions are different, but they show dramatic different activity. Thus, additional structural analyses or mutagenesis studies are desired to show why these two super-substrates are differently recognized NSD2 and SETD2. Guided by these information, authors could identify NSD2-biased and SETD2-biased non-histone substrates, which will be of broad interest to the readers of Communication Biology.”

Reply: Thank you for raising this very interesting question. However, the similarity of the two peptide interfaces of NSD2 and SETD2 is much lower than one might anticipate. Below a part of the multiple sequence alignment of NSD enzymes and SETD2 is shown with the residues highlighted that were shown to be involved in a differential interaction with the H3K36 and ssK36 substrates in NSD2 (blue shade, data from the current manuscript) or SETD2 (red shade, data are taken from Schuhmacher et al., 2020). Strikingly, only 12 out of the 42 NSD2 residues (29%) are identical between NSD2 and SETD2 indicating that the peptide interaction is very different in its molecular details in both cases.

This divergence more specifically manifests in the peptide interaction of both enzymes at the 3 residues differing between both super-substrates: at the -4 site (R vs. K), at the -3 site (T vs. F), and at

the +4 site (N vs. R). The SETD2 and NSD2 residues preferentially interacting with these substrate residues are listed in the following table.

Pos.	SETD2		NSD2	
	ssK36 sequence	interacting residues	ssK365 sequence	interacting residues
-4	R	E1674	K	I1114 (not conserved) T1115 (not conserved) N1186 (not conserved) E1187 K1188 (not conserved) T1189 (not conserved)
-3	F	E1674 A1675 (not conserved) Q1676 (not conserved)	T	T1115(not conserved) H1116 F1117 (not conserved) L1181 (not conserved) C1183 (not conserved) K1188 (not conserved) T1189 (not conserved)
+4	R	M1526 Q1638 (not conserved)	N	N1178 (not conserved)

Evidently, from the 20 amino acids involved in the recognition of these 3 substrate residues in both enzymes, only one is involved in a comparable interaction in both cases (E1674/E1187 interacting with the -4 residue), but even this residue is engaged in different contact patterns to the -3 substrate residue. Hence, depicting the molecular differences between the substrate interaction of NSD2 and SETD2 super-substrates will be a highly complicated task, that certainly is beyond the scope of what can be done in a revision of this work.

“3. To identify novel substrates of NSD2, authors used ScanSite server to search with a sequence motif which does not strictly follow super-substrate sequence. As a result, the identified sequences from database (e.g. ATRX and FANCM) are far different from the super-substrate sequence (as shown below). I wonder whether there is a sequence nearly-perfectly matching the super-substrate sequence (probably just one or two position varied) and whether this ssK36-like sequence could be methylated more efficiently by NSD2 than ATRX or FANCM.

ATRX: CHFPGIKQIKNGTT

FANCM:HKKSSFIKNINQGSS

ssK36:APKTGGVCRPNNYRP

H3K36:APATGGVKKPHRYRP”

Reply: Scansite does not allow to search for a pattern with one or two ambiguities. However, as the super-substrate matches the general specificity profile used in our ScanSite search, all such sequences were retrieved and included in the analysis.

“4. Authors claimed that H3K36 is methylated at least 100-fold weaker than super-substrate ssK36. This estimation might be too rough. Is that possible to measure the kinetic parameters for NSD2-catalyzed reaction to get a more accurate number?”

Reply: Thank you for highlighting this point. We have repeated the methylation analyses using ssK36 and H3K36 peptides as well as the GST-tagged H3K36 and ssK36 protein substrates in new experiments now shown in Figure 2D and E. Based on dilution series of the methylated ssK36 peptide or protein loaded on the same gel together with the corresponding H3K36 sample, we can now determine that ssK36 is methylated 91-fold more readily at the peptide level and about 3000-fold better at protein level than H3K36. This point has been explained in the text now on p. 6-7. The description of the fold-stimulation has been adjusted correspondingly throughout.

REVIEWERS' COMMENTS:

Reviewer #1 (Remarks to the Author):

The authors have adequately responded to all the comments and suggestions from the first round of review. I recommend publication and have one minor comment.

However, I respectfully disagree with their reply to point #9: "These are just autoradiographic images of gels not showing substrate methylation signals. All of them show NSD2 automethylation as a kind of "positive control" ensuring the technical correctness. All experiments were conducted at least twice. We do not think that it will be helpful for readers to show all negative images."

It is important for readers to observe the quality of the protein preparation for these 15 in order to interpret it as a true negative.

Reviewer #2 (Remarks to the Author):

In the current revision the authors have addressed all of my concerns, I can recommend the manuscript for publication now.

Reviewer #3 (Remarks to the Author):

In this revised version of their manuscript, Jeltsch and coworkers addressed the issues raised by the reviewers after the first submission and consequently further improved the manuscript. In particular, they included additional details, updated figures and tables and commented accordingly the results obtained.

Also, I appreciated the answers and explanations they provided in their point-by-point reply. Therefore, in my opinion, now the manuscript perfectly fits the aims of Communications Biology, and I am glad to recommend it for publication.

Reviewer #4 (Remarks to the Author):

The authors have addressed most of the criticisms and performed several suggested experiments to address critical points. Overall, I am satisfied with the revisions and feel that the quality of the revised manuscript has been improved.